# Ultrastable microwave and soliton-pulse generation from fibre-photonic-stabilized microcombs

Dohyeon Kwon[1,4], Dongin Jeong[2,4], Igju Jeon[1], Hansuek Lee [2,3] & Jungwon Kim [1]

The ability to generate lower-noise microwaves has greatly advanced high-speed, high-precision scientific and engineering fields. Microcombs have high potential for generating such low-noise microwaves from chip-scale devices. To realize an ultralow-noise performance over a wider Fourier frequency range and longer time scale, which is required for many high-precision applications, free-running microcombs must be locked to more stable reference sources. However, ultrastable reference sources, particularly optical cavity-based methods, are generally bulky, alignment-sensitive and expensive, and therefore forfeit the benefits of using chip-scale microcombs. Here, we realize compact and low-phase-noise microwave and soliton pulse generation by combining a silica-microcomb (with few-mm diameter) with a fibre-photonic-based timing reference (with few-cm diameter). An ultrastable 22-GHz microwave is generated with $-110$ dBc/Hz ($-88$ dBc/Hz) phase noise at 1-kHz (100-Hz) Fourier frequency and $10^{-13}$-level frequency instability within 1-s. This work shows the potential of fully packaged, palm-sized or smaller systems for generating both ultrastable soliton pulse trains and microwaves, thereby facilitating a wide range of field applications involving ultrahigh-stability microcombs.

[1] School of Mechanical and Aerospace Engineering, Korea Advanced Institute of Science and Technology (KAIST), Daejeon 34141, Korea. [2] Graduate School of Nanoscience and Technology, Korea Advanced Institute of Science and Technology (KAIST), Daejeon 34141, Korea. [3] Department of Physics, Korea Advanced Institute of Science and Technology (KAIST), Daejeon 34141, Korea. [4]These authors contributed equally: Dohyeon Kwon, Dongin Jeong. ✉email: hansuek@kaist.ac.kr; jungwon.kim@kaist.ac.kr

The generation of lower-noise and higher-stability micro-waves has advanced a wide range of fields in science and engineering ranging from radio astronomy and particle accelerators to radars and telecommunications. Various photonic methods have been developed to generate such low-noise microwaves in the last three decades[1–9], for example, optoelec-tronic oscillators (OEOs)[1,2], electro-optical frequency division[3–5] and monolithic mode-locked lasers[6]. In particular, by coherently linking the optical and microwave frequency domains, the use of optical frequency combs has become a powerful method for producing low-noise and ultrastable microwave signals[7–10]. The optical frequency division (OFD) of an ultrastable continuous-wave (CW) light by a mode-locked fibre frequency comb has enabled the generation of −130 dBc/Hz-level microwaves at 10-Hz Fourier frequency[10]. In addition to low-noise microwave generation, frequency combs and photonic methods are also highly effective at transferring low-noise and synchronised microwave signals and optical pulse trains to remote locations for newly emerging ultrahigh-precision applications, for example, the synchronisation of large-scale X-ray science facilities[11–13] and the distribution of phase-synchronised microwaves for very long baseline interferometry (VLBI)-based radio astronomy[14,15].

More recently, microresonator-based optical frequency combs (microcombs), different from traditional mode-locked oscillator-based combs, have been intensively studied and rapidly developed[16–22]. Microcombs generate soliton optical pulse trains using a nonlinear frequency conversion process of parametric four-wave mixing (FWM) in a microresonator. The main advantages of microcombs, in contrast to mode-locked oscillator combs, are their compact size and high repetition rate. Micro-combs can generate soliton-pulse trains with ~10 GHz to ~1 THz repetition rate from tens-of-micrometre to few-millimetre-sized chips. Further, the quantum-limited fast timing jitter of free-running microcombs[23] was reported to be on the order of only a few femtoseconds[24,25], the regime of which is difficult to achieve with electronic oscillators. These features make micro-combs an ideal comb source for various microwave photonic applications[26–30]. Indeed, recent works have shown the potential for microcomb-based microwave generation[31–36]. For example, a free-running 14-GHz MgF$_2$ microcomb reported the phase noise of −115 dBc/Hz (−80 dBc/Hz) at 1-kHz (100-Hz) Fourier frequency[32] and a free-running 20-GHz silicon-nitride micro-comb demonstrated the phase noise of −80 dBc/Hz (−45 dBc/Hz) at 1-kHz (100-Hz) Fourier frequency[33].

Although free-running microcombs have demonstrated the potential to reach a previously inaccessible regime of ultralow timing jitter and microwave phase noise with repetition rates extending to tens of gigahertz, timing jitter and repetition-rate phase noise must be suppressed over a longer time scale (down to lower Fourier frequency range) to satisfy a wide range of high-precision applications. Accordingly, several previous works have attempted to stabilise the microwave repetition rate of micro-combs, including microwave phase-locking using frequency synthesisers[32–35,37,38] and ultrastable cavity-locked pump lasers[34,39]. However, the lack of a compact, mechanically robust, broadband and high-performance microcomb stabilisation method has been a major bottleneck in achieving the field applications of ultrastable and compact microcombs in micro-wave photonic systems.

Here, we demonstrate the generation of low-phase-noise K-band microwaves from a palm-sized photonic platform by combining a silica microcomb and a fibre-photonic timing sta-bilisation method. To stabilise the timing, a kilometres-long fibre delayline, wound as a compact spool with a several-cm outer diameter, which was previously used for stabilising mode-locked Er-fibre combs[40], was used as a timing reference. In order to extend the repetition-rate locking bandwidth over >100 kHz, we also developed a new high-speed repetition-rate tuning mechanism using a customised voltage-controlled oscillator (VCO) and an acousto-optic frequency shifter (AOFS), which can be also applied for other types of ultrahigh-Q resonator-based combs. In addition to the compact size of the platform, this stabilisation method operates at room temperature, does not require a vacuum environment and is completely alignment-free, making it ideal for use with ultracompact microcombs. By employing the fibre-photonic stabilisation, the 22-GHz micro-wave phase noise generated from a free-running silica microcomb (−30 dBc/Hz at 10 Hz Fourier frequency) could be suppressed by >40 dB down to −74 dBc/Hz. The resulting integrated timing jitter of the 22-GHz microwave signal is only 10.4 fs when integrated from 10 Hz to 1 MHz, which corresponds to the $10^{-13}$-level frequency instability within 1-s. With its excellent phase noise, stability, compactness and robustness, the demonstrated system can be readily applied in various field applications, including radio astronomy, photonics-based radars[41], 5/6 G tel-ecommunications and signal analysis instruments.

## Results

**System overview**. The principle of ultrastable microwave and soliton-pulse generation from a timing-stabilised silica micro-comb is shown in Fig. 1a. A pulse train with 22-GHz repetition rate is generated via FWM and soliton mode-locking processes in a silica microresonator with a 3-mm-diameter driven by a CW pump laser. A 22-GHz microwave signal is extracted from an optical pulse train by an optical-to-electronic (OE) conversion process using a modified uni-travelling carrier (MUTC) photo-diode followed by a radio-frequency (RF) bandpass filter and an RF amplifier. Since the timing jitter and repetition-rate phase noise from a free-running microcomb diverge over time, the repetition rate is stabilised by a fibre-photonic stabiliser[42], which obtains signals containing the frequency noise of two comb modes (i.e. $v_m = mf_{rep} + f_{ceo}$ and $v_n = nf_{rep} + f_{ceo}$, where $f_{rep}$ is the repetition rate, $f_{ceo}$ is the carrier-envelope offset frequency and m and n are mode numbers; see Fig. 1b). The frequency noise dif-ference between two comb modes is extracted by mixing the signals, which corresponds to the frequency noise of $(m-n)f_{rep}$. Once the feedback loop is closed, the stability of the microcomb repetition rate ($\delta f_{rep}/f_{rep}$) can reach up to the stability of the time delay of the fibre link ($\delta\tau/\tau$).

**Soliton microcomb**. A silica wedge microresonator is used for the soliton generation. The free spectral range ($D_1/2\pi$) of a 3-mm-diameter resonator is ~22 GHz. The intrinsic Q factor of the resonator is $2.5 \times 10^8$ and the second-order dispersion ($D_2/2\pi$) is 13.3 kHz. A pump laser at 1550 nm is amplified by an Er-doped fibre amplifier (EDFA; EDFA$_1$ in Fig. 1a) and the amplified spontaneous emission (ASE) noise is removed by a bandpass filter. The pump laser is coupled into and out of the resonator using a tapered fibre after the polarisation is adjusted to the microresonator mode. A rapid scan of the pump frequency from blue-to-red detuning allows soliton generation. To avoid thermal destabilization of the soliton, the pump-cavity detuning is locked via a piezoelectric transducer (PZT) of the pump laser[43,44]. An acousto-optic frequency shifter (AOFS; AOFS$_1$ in Fig. 1a) miti-gates thermal destabilization by implementing rapid power modulation within a few microseconds when a soliton is gener-ated and allows frequency modulation to stabilise the repetition rate (see Methods and Supplementary Information). The residual pump signal is removed by a notch filter and the soliton is amplified by an EDFA (EDFA$_2$ in Fig. 1a) for microwave gen-eration and comb stabilisation. From the amplified power, 90

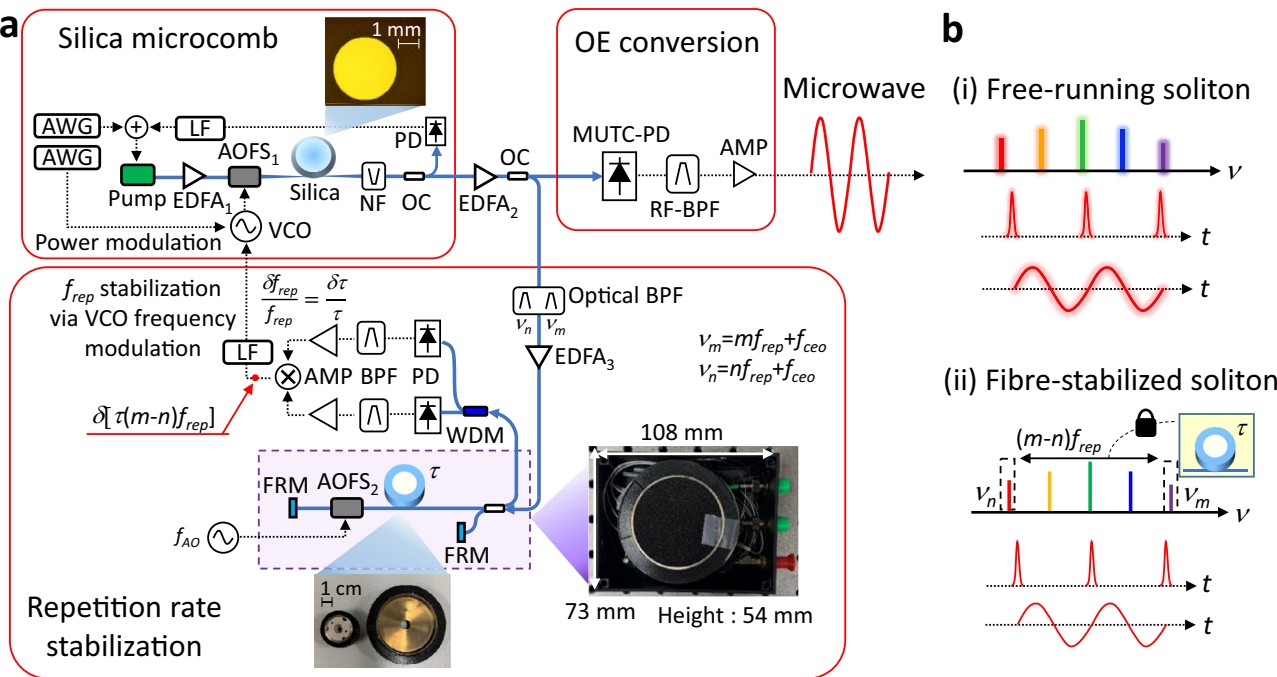

**Fig. 1 Ultrastable microwave generation from a silica microcomb stabilised by fibre photonics. a** Schematic of ultrastable microwave generation from a fibre-stabilised silica microcomb. AWG arbitrary waveform generator, EDFA erbium-doped fibre amplifier, AOFS acousto-optic frequency shifter, VCO voltage-controlled oscillator, NF notch filter, OC optical coupler, PD photodiode, LF loop filter, MUTC-PD modified uni-travelling carrier photodiode, RF-BPF 250-MHz-bandwidth radio-frequency bandpass filter at 22-GHz, AMP RF amplifier, BPF RF-bandpass filter at $2f_{AO}$, Optical BPF bandpass filter for $\nu_n$ and $\nu_m$, $f_{rep}$ repetition rate, $f_{ceo}$ carrier-envelope offset frequency, $m$ and $n$ mode number of $\nu_m$ and $\nu_n$, respectively, FRM Faraday rotating mirror, $f_{AO}$ driving frequency for AOFS$_2$, $\tau$ time delay induced by the fibre link. An optical microscope image of a silica microresonator is shown at the top of the figure. A photograph of a 200-m-long fibre link (left) and a 1-km-long fibre link (right) is displayed at the bottom of the figure. A photograph of the entire fibre-photonic repetition-rate stabiliser (with a case size of 108 mm × 73 mm × 54 mm) is shown on the right side of the figure. **b** Concept of comb-mode frequency noise, timing jitter, microwave phase noise of a free-running microcomb (i) and a fibre-stabilised microcomb (ii).

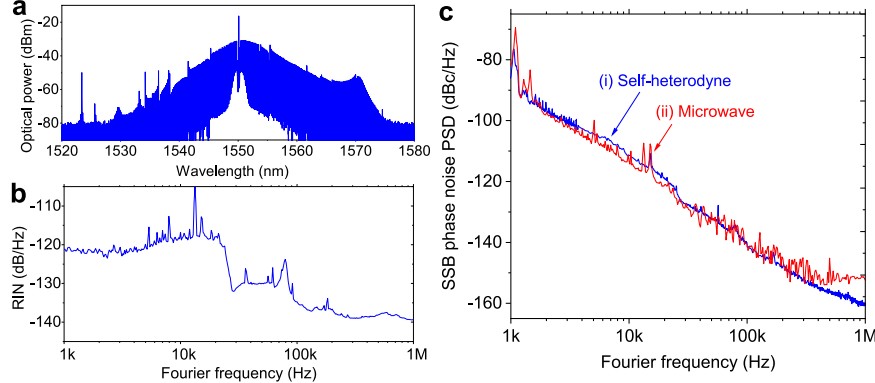

**Fig. 2 Properties of the free-running silica microcomb. a** Optical spectrum of the silica microcomb. **b** Relative intensity noise (RIN) of the microcomb. Note that the sudden jump at 20 kHz is due to the response of the PZT in the pump laser. **c** Repetition-rate phase noise of the free-running microcomb measured in the (i) optical domain and (ii) microwave domain. Source data for Fig. 2a, b and c are provided in the Supplementary Information.

percent of the amplified power is used to generate microwaves, while 10 percent of the optical power is tapped to stabilise the repetition rate.

The typical performance of the soliton is shown in Fig. 2. The estimated pulsewidth from the sech²-shaped optical spectrum of the microcomb (Fig. 2a) is 350 fs. The intensity fluctuation of the pulse train is measured as the relative intensity noise (RIN), as shown in Fig. 2b. The single-sideband (SSB) repetition-rate phase-noise power spectral density (PSD) of the free-running microcomb (curve (i) in Fig. 2c) is first measured in the optical domain by a fibre delayline-based self-heterodyne method[24,45] (see Supplementary Fig. 10). Due to the high-resolution of

the self-heterodyne measurement method, the repetition-rate phase noise is accurately characterised below −160 dBc/Hz at 1-MHz Fourier frequency. We also measured the SSB phase noise of the extracted 22-GHz microwave signal (curve (ii) in Fig. 2c) using a commercial phase noise analyser (Keysight, E5052B + E5053A), which shows a good agreement with the optical-domain measurement result (curve (i) in Fig. 2c). The slight difference in the >500-kHz Fourier frequency range is due to the photodetection white noise floor and the measurement sensitivity of the phase noise analyser.

The repetition-rate phase noise of the free-running comb can be minimised by adjusting the pump-cavity frequency detuning

to the quiet point[31,33] (see Supplementary Fig. 3). In our system, the phase noise is minimised when the pump-cavity detuning ($\delta\omega$) is 16 MHz according to the relation $\delta\omega = (D_2/2D_1{}^2)(1/\tau_p{}^2)$, where $\tau_p$ is the pulsewidth[31]. The measured phase noise at 10-kHz (100-kHz) Fourier frequency is −111 dBc/Hz (−141 dBc/Hz) with an integrated timing jitter of 2.14 fs (249 as) when integrated from 10-kHz (100-kHz) to 1-MHz Fourier frequency range. Compared to the recently demonstrated free-running 20-GHz silicon-nitride microcomb[33], the phase noise of the demonstrated silica microcomb is more than 10 dB lower at 100-kHz Fourier frequency due to the material properties of the silica microresonator[23,31,46].

Note that we conducted the repetition-rate stabilisation where the pump-cavity detuning is ~500-kHz blue side to the quiet point, where the lowest phase noise of the stabilised microcomb is achieved. We believe that, at the exact quiet point, the comb is the least sensitive to the pump disturbance (i.e. $\delta f_{rep}/\delta\nu_{pump} \sim 0$) and has a difficulty in increasing the loop gain when stabilising[34]. We found that the repetition-rate phase noise (timing jitter) could be suppressed most at a slightly blue-detuned position from the exact quiet point, where the free-running jitter is still low and the pump sensitivity is high enough for effectively increasing the loop gain of the feedback control loop.

**Microcomb repetition-rate stabilisation.** The working principle of low-phase-noise microwave generation is based on repetition-rate stabilisation using a compact fibre Michelson interferometer, as illustrated in Fig. 1a (also see Supplementary Figs. 10 and 11). Two frequency modes (i.e. $\nu_m = mf_{rep} + f_{ceo}$ and $\nu_n = nf_{rep} + f_{ceo}$) are filtered by fibre-Bragg gratings to extract repetition-rate frequency noise. A fibre link is used as both the timing reference and the frequency noise discriminator. In this work, we selected a 1-km-long (2-km-long round-trip) fibre link by considering both detection sensitivity (scaling with delay time $\tau$) and detection bandwidth (scaling with inverse of delay time, $1/\tau$). Note that, depending on the intended performance and application, different length of fibre link can be also used (see Supplementary Fig. 8 for comparison of stabilised phase noise with different delay length). The fibre link is wound with few-centimetre diameter

(e.g. 7 cm diameter for 1-km-long fibre) using the fibre-winding method used for making fibre-optic gyroscopes. An AOFS driven by $f_{AO}$ (AOFS$_2$ in Fig. 1a) is inserted at the delay arm to reduce background noise by synchronous detection. The absolute frequency noise of each comb mode by optical carrier interference is photodetected individually using a wavelength division multiplexing (WDM) coupler. The carrier frequency of each photodetected signal is $2f_{AO}$ induced by the round-trip frequency shift due to AOFS$_2$ in the delay arm. The phase noise of $2f_{AO}$ contains the absolute frequency noise of each corresponding comb mode weighted by the time delay (i.e. $\delta[\tau \times (mf_{rep} + f_{ceo} + 2f_{AO})]$ and $\delta[\tau \times (nf_{rep} + f_{ceo} + 2f_{AO})])$). Each photodetected signal is bandpass-filtered at $2f_{AO}$ and mixed by a frequency mixer, which results in the rejection of common-mode $f_{ceo}$ and $2f_{AO}$ noise. The resulting lowpass-filtered baseband mixer output, therefore, contains only the repetition-rate frequency noise (i.e. $\delta[\tau \times (m-n)f_{rep}]$).

While mode-locked fibre combs could be easily locked to the fibre reference by intra-cavity PZTs, locking the microcombs, ultrahigh-Q resonator-based combs such as silica microcombs in particular, is not straightforward due to the lack of a direct mechanical actuator inside the microresonator. In particular, the previous approaches using pump power modulation could achieve only a few kHz bandwidth[47], which is too narrowband for effective repetition-rate stabilisation. In this work, we employed the AOFS for broadband locking of silica microcombs to the fibre reference. Since the repetition-rate frequency noise is coupled to the residual cavity-pump detuning noise[31], it can be suppressed via feedback control to the voltage-controlled oscillator (VCO) applied to the AOFS before the silica microresonator (AOFS$_1$ in Fig. 1a). Here, the frequency modulation-based stabilisation can provide a much larger control bandwidth than pump power modulation: the 3-dB control bandwidth of the demonstrated system can reach ~120 kHz as shown on the curve (ii) in Fig. 3a. Note that this control bandwidth was limited by the fibre delay time itself (which is set by $1/\tau$), and the actuation bandwidth itself achieved by the AOFS and the customised VCO is much larger, over 400 kHz. Also note that, while this broadband repetition-rate tuning mechanism is

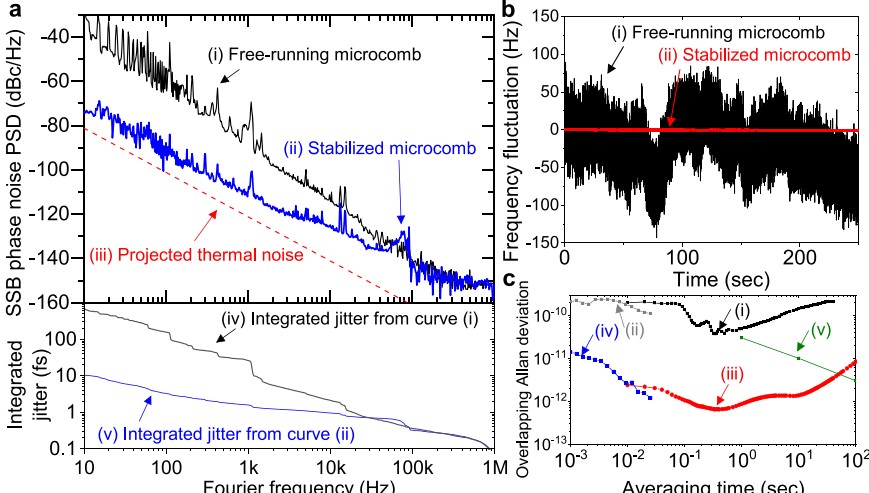

**Fig. 3 Characterisation of the fibre-stabilised silica microcomb. a** 22-GHz microwave phase noise and integrated timing jitter. (i) Phase noise of the free-running microcomb. (ii) Phase noise of the 1-km fibre-stabilised microcomb. (iii) Projected thermal noise-limited phase noise[53] of the 1-km fibre-photonic stabiliser. (iv) Integrated root-mean-square timing jitter of the free-running microcomb. (v) Integrated root-mean-square timing jitter from the fibre-stabilised microcomb. **b** Long-term frequency fluctuation of the repetition rate. (i) Free-running microcomb (black). (ii) Fibre-stabilised microcomb (red). **c** Repetition-rate frequency instability of the microcomb. (i) Free-running microcomb (computed from curve (i) in Fig. 3b). (ii) Free-running microcomb (computed from curve (i) in Fig. 3a). (iii) Fibre-stabilised microcomb (computed from curve (ii) in Fig. 3b). (iv) Fibre-stabilised microcomb (computed from curve (ii) in Fig. 3a). (v) Chip-scale atomic reference (SA55, Microsemi). Source data for Fig. 3a, b and c are provided in the Supplementary Information.

applied for a silica microcomb and a fibre delayline-based reference source in our work, it can be also used for other types of ultrahigh-Q microresonators and reference sources for broadband stabilisation of microcomb repetition rates. We also confirmed that the crosstalk between the repetition-rate stabilisation via frequency modulation of the VCO and the soliton mode-locking via the pump laser PZT is negligible, and both controls can be used simultaneously (see Supplementary Information and Supplementary Fig. 7).

For the demonstrated system, a fibre Michelson interferometer using a 1-km-long fibre link with a diameter of 7 cm was packaged in a 108 mm × 73 mm × 54 mm-sized, air-tight box (see Fig. 1a). Note that, the present-day size of the fibre-photonic stabiliser is mostly limited by the size of AOFS$_2$ due to the required space for the frequency-shifted diffracted beam inside an AOFS device. The system size can be further reduced by using an SSB phase modulator[47,48] instead of an AOFS. In addition, fibre-optic components can be compactly integrated on a single photonic integrated circuit (PIC) assisted by silicon photonics[26,49–51].

**OE conversion.** To avoid a pulsewidth-dependent noise floor at high offset frequencies[52], the soliton pulsewidth is compressed to <1 ps using a dispersion compensating fibre. More than 15 mW of optical power is applied to an MUTC photodiode. The generated photocurrent with a −8 V bias voltage is ~7 mA. The generated microwave is filtered by a bandpass filter centred at 22 GHz and amplified by a low-phase-noise RF amplifier up to +10 dBm.

**Microwave phase noise and frequency stability performances.** The phase noise of the 22-GHz microwaves extracted from the silica microcomb is characterised by commercial phase noise analysers (Keysight, E5052B + E5053A). The SSB PSD of the 1-km-long fibre-stabilised microcomb is shown as curve (ii) in Fig. 3a. The phase noise PSD of the fibre-stabilised microcomb is −110 dBc/Hz (−88 dBc/Hz) at 1-kHz (100-Hz) Fourier frequency. Compared to that of the free-running microcomb (curve (i) in Fig. 3a), the phase noise of the fibre-stabilised microcomb (curve (ii) in Fig. 3a) is suppressed by more than 30 dB for Fourier frequency of <100 Hz, which results in effective frequency stabilisation of the microcomb-generated microwave signal. Using the fibre stabiliser, the integrated root-mean-square (RMS) timing jitter is greatly suppressed from 689 fs (curve (iv) in Fig. 3a) to 10.4 fs (curve (v) in Fig. 3a) over the 0.1-s integration time. The timing jitter over 0.1-ms time scale, which is important for telecommunications and analogue-to-digital data converters (ADCs), reaches even the subfemtosecond regime (0.96 fs) [integration bandwidth: 10 kHz–1 MHz]. Note that, in the high Fourier frequency (>300 kHz), the measured photodetection noise floor (~−153 dBc/Hz) is degraded from the projected shot noise limit (−161 dBc/Hz) by the noise figure as well as the amplitude-to-phase conversion of the used RF amplifier.

The fundamental phase-noise limit of the fibre-photonic stabiliser originates from the thermomechanical and thermo-conductive fibre-length fluctuation[53] that destabilises the timing stability of the fibre delayline (curve (iii) in Fig. 3a, see Supplementary Information). There is a ~10-dB discrepancy between the stabilised phase noise and the fundamental limit; we believe that this discrepancy originates from the Rayleigh scattering-limited intensity noise to the frequency noise conversion[54] and the limited detection sensitivity of the fibre stabiliser, where the error signal, $\delta[\tau \times (m - n)f_{rep}]$, scales with the fibre delay time $\tau$ and the frequency separation between bandpass filters $(m - n)f_{rep}$ (see Supplementary Information).

The long-term stability is evaluated by the frequency instability computed from the frequency fluctuation. To measure the frequency fluctuation with high-sensitivity digital cross-correlation[55], the K-band microwave is down-converted to a low frequency. Two independent silica microcombs, each stabilised to an independent fibre-photonic stabiliser, are used for the down-conversion (see Supplementary Information and Supplementary Fig. 12). The fundamental repetition rates of the two microcombs are 22.059 and 22.083 GHz, and the down-converted microwave beat note of the two microcombs is 23.3 MHz. The frequency fluctuation of the microwave beat note is sampled by a frequency counter with a sampling rate of 100 Hz. While the frequency fluctuation of the free-running microcombs (curve (i) in Fig. 3b) is larger than 100 Hz, the frequency fluctuation of the fibre-stabilised microcombs (curve (ii) in Fig. 3b) is less than 2 Hz. The frequency instability evaluated by the overlapping Allan deviation is computed from the frequency fluctuation measurement (for >10$^{-2}$ s) and the phase noise measurement[56] (from 10$^{-3}$ s to 10$^{-2}$ s). For the free-running microcombs (curves (i) and (ii) in Fig. 3c), the minimum frequency instability is on the level of 10$^{-11}$ and diverges after 0.4 s. For the fibre-stabilised microcombs (curves (iii) and (iv) in Fig. 3c), the frequency instability starts from $1.4 \times 10^{-11}$ at 10$^{-3}$ s and reaches $4.6 \times 10^{-13}$ at 0.4 s.

The frequency instability starts increasing after 0.4 s due to the time-dependent drift. For the free-running microcomb, we believe that the time-dependent timing drift at this time scale is due to the thermally-induced path length change of the silica material, which was also observed in silica-based cavity mirrors[57]. For the fibre-stabilised microcomb, the stability of the repetition rate follows the timing stability of the fibre link. The SMF-28 fibre has a thermal coefficient of delay (TCD) of ~40–130 ps/km/K[58,59], and this time-dependent temperature drift in the fibre link induces Allan deviation diverging after ~0.4 s (see Supplementary Information). To ensure longer-term stable operation while retaining the compact overall size, chip-scale atomic references can be additionally used. For example, one of the best chip-scale atomic references (curve (v) in Fig. 3c, SA55 from Microsemi) can improve the performance of our system after 40 s, where the frequency instability of the atomic reference becomes lower than that of the fibre-stabilised microcomb (~$3 \times 10^{-12}$ at 40 s).

Finally, the microwave phase noise of the fibre-stabilised microcomb is compared with that of previous microcombs (operating in the X-, Ku- and K-bands) in Fig. 4, where all phase noise PSDs are scaled to the same 22-GHz carrier frequency.

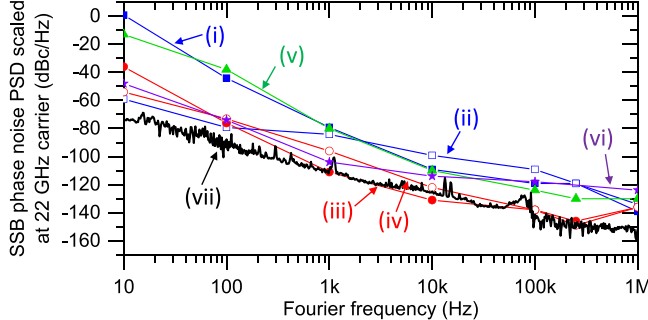

**Fig. 4 Phase noise performance comparison.** Phase noise comparison with other microcomb-based microwaves scaled to 22 GHz. (i) Free-running 20-GHz Si$_3$N$_4$ microcomb[33], (ii) Stabilised 20-GHz Si$_3$N$_4$ microcomb[33], (iii) Free-running 14-GHz MgF$_2$ microcomb[32], (iv) Stabilised 14-GHz MgF$_2$ microcomb[32], (v) Free-running 22-GHz SiO$_2$ microcomb[31], (vi) Free-running 10-GHz MgF$_2$ microcomb[60]. (vii) Fibre-stabilised 22-GHz SiO$_2$ microcomb (this work). Source data for Fig. 4 are provided in the Supplementary Information.

Overall, our result exhibits lower phase noise compared to other material platforms (bulk crystals, silica and silicon nitrides)[31–33,60] as well as microwave source-stabilised combs[32,33].

## Discussion

In summary, we have generated both a low-jitter soliton-pulse train and low-noise K-band microwave signal from a compact photonic platform by combining an ultralow-jitter silica microcomb, an AOFS-based broadband repetition-rate tuning mechanism and an ultrastable fibre delayline. Previously, low-phase-noise microwaves over millisecond time scales from microcombs were obtainable only by using bulky external reference sources such as RF signal generators[32–35,37,38,47], electro-optic combs[61,62], cavity-locked CW lasers[34,39] and fully locked optical frequency combs[33,63]. In this work, we showed that the phase noise and frequency fluctuation of the demonstrated silica microcomb can be effectively suppressed by using fibre-optic components fully packaged in a palm-sized enclosure without a vacuum environment. Due to its compact size and alignment-free operation, the demonstrated system will greatly advance low-noise photonic microwave generators for field applications, including photonics-based radars[41], 5 G/6 G telecommunications, software-defined radios[64], signal analysis and VLBI[14,15]: as a specific example, photonics-based radars[41], which combine optical pulse-based sampling/ ADC and low-noise microwave generation, can be greatly benefited by having both high repetition-rate (>20 GHz) low-jitter optical pulse trains and very low-phase noise K-band microwave signals generated from a compact platform. The demonstrated fibre-photonic stabiliser is also readily applicable to the stabilisation of other microcombs with different material platforms[59,65–73] and repetition rates reaching even the terahertz range[26,62,74]. We also believe that the system has high potential to become even more compact to the chip scale by combining it with silicon photonics. For example, fibre-Bragg gratings and WDM couplers can be replaced with the chip-scale WDMs, and a fibre link can be replaced with a silicon chip[75] or a microresonator[76].

## Methods

**Resonator fabrication**. Silica wedge resonators with a radius of 3 mm are fabricated through wet oxidation (Tystar, Tytan Mini 1800), optical lithography (SUSS Micro Tec, MA-6 aligner), wet etching (J.T. Baker, buffered oxide etchant 6:1) and dry etching (Teraleader, XeF$_2$ etching system), as detailed in ref. [77]. A high-purity float-zone silicon wafer with an 8-μm oxide film is used. The difference in repetition rate is implemented as a difference in the resonator size (several micrometres). Since both resonators are fabricated at the same time and share the same geometric structure, the wedge angle is ~16° and the second-order dispersion parameter $D_2/2\pi$ is 13.3 kHz. The intrinsic Q factor of both resonators is 250 million.

**Soliton mode-locking and microcomb**. The soliton mode is locked using the power kicking method and active capture technique[43,44] (Supplementary Fig. 1). A narrow linewidth CW laser is used as the pump laser. The pumping power is ~120 mW. The pump is coupled to a microresonator by a tapered fibre, where the polarisation is adjusted to the microresonator by a polarisation controller. First, the laser frequency is swept from the blue side with a sweep rate of ~1 MHz/μs using an arbitrary waveform generator (AWG). As the coupling power to the soliton mode increases, a frequency comb is formed via FWM beyond the threshold for parametric oscillation. After the frequency sweep near the cavity resonance, the pump power is quickly modulated (it decreases for a few μs and then increases for 50–150 μs) using an AOFS$_1$, and the soliton is triggered. AOFS$_1$ is driven by a VCO whose working range is 51–59 MHz. A laser servo is engaged after the soliton is triggered, and the laser frequency control via the laser PZT achieves the long-term operation of the soliton. Here, the photodetected soliton power is used for the servo error.

**High-speed repetition-rate tuning mechanism**. In our experiment, AOFS$_1$ in Supplementary Fig. 5 is used for both power kicking and stabilisation. To implement both functions using AOFS$_1$, we customised the voltage-controlled oscillator

(VCO) to modulate the power and frequency. The power kicking and the repetition-rate stabilisation are conducted through the power modulation port and the frequency modulation port of the VCO, respectively. As the frequency of the pump laser is already locked to the specific soliton mode via PZT, we applied the frequency modulation of VCO for the repetition-rate stabilisation. We could implement the stabilisation based on the frequency modulation for broadband feedback control loop over 100 kHz bandwidth.

**Fibre delayline-based repetition-rate stabilisation method**. For repetition-rate phase noise stabilisation, a Michelson interferometer is introduced (Type 2 in Supplementary Fig. 10). The time delay ($\tau$) is introduced by a fibre link, which determines the stability of the fibre reference system ($\delta\tau/\tau$), the frequency discrimination sensitivity (scaling with $\tau$) and the achievable locking bandwidth (scaling with $1/\tau$). A compact 1-km-long fibre (diameter 70 mm, height 18 mm) is used for the main experiment. Each arm of the fibre Michelson interferometer is split by a fibre coupler. Faraday rotating mirrors are used as end mirrors of the interferometer. An AOFS$_2$ driven by $f_{AO}$ is inserted to perform the synchronous detection to reduce background noise. At the interferometer output, each frequency mode is filtered by a WDM and photodetected. The phase noise of the photodetected signal at $2f_{AO}$ (due to the round-trip in the interferometer) contains the frequency noise of each filtered mode weighted by the time delay, in the form of $\delta[\tau \times (mf_{rep} + f_{ceo} + 2f_{AO})]$ and $\delta[\tau \times (nf_{rep} + f_{ceo} + 2f_{AO})]$. Each photodetected signal is filtered by an RF-bandpass filter at $2f_{AO}$, amplified and down-converted to the baseband by a frequency mixer. The lowpass-filtered baseband mixer output contains the frequency noise of $\delta[\tau \times (m-n)f_{rep}]$ by rejecting common-mode $f_{ceo}$ noise and $2f_{AO}$ noise. Note that, in this experiment, $(m-n)f_{rep}$ corresponds to ~2.5 THz due to the frequency difference between $mf_{rep} + f_{ceo}$ and $nf_{rep} + f_{ceo}$. Since the pump laser PZT is already used to prevent the soliton from experiencing thermal destabilization, the phase noise is suppressed via the extra frequency modulation of the pump frequency using the VCO applied to the AOFS (AOFS$_1$ in Fig. 1 and Supplementary Fig. 5).

## Data availability

Source data are provided with this paper.

## Code availability

The simulation and computational codes for this study are available from the corresponding authors on request.

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

## Acknowledgements

This research was supported by the Institute for Information and Communications Technology Promotion of Korea (Grant no. 2019-0-01349 for JK and HL) and the National Research Foundation of Korea (Grant no. 2021R1A2B5B03001407 for JK). We thank Minji Hyun for the discussions. We also thank the Center for Time and Frequency, Korea Research Institute of Standards and Science (KRISS), for loaning the phase noise analyser.

## Author contributions

D.K. and J.K. conceived the idea and designed the experiment. J.K. and H.L. supervised the project. D.J. and H.L. designed and fabricated the silica microresonator. D.K., D.J. and I.J. performed the experiment and obtained data. D.K. and J.K. analysed the data and wrote the manuscript with inputs from all authors.

## Competing interests

The authors declare no competing interests.

**Additional information**

