## [Peer Review File · Nature Communications]

Reviewers' Comments:

Reviewer #1:

Remarks to the Author:

The manuscript reports a method for stabilizing repetition rate of soliton microcombs using a compact fiber-based timing reference. Such method allows detection and stabilization of the relative phase noise between two distantly located comb lines, which features orders of magnitude better sensitivity than using the detected soliton beatnote as an indicator. As a result, ultrastable 22-GHz microwave signal is generated from a chip-based silica microresonator, with more than 30 dB reduction in phase noise at low offset frequencies compared with free-running case. As a remark, the authors also compare two independent stabilized soliton microcombs to accurately reveal the improved long-term stability.

This work appears to be of high quality and has done adequate work to demonstrate the performance of this approach. The elaboration of key principles is straightforward and is well supported by detailed experiments and convincing analysis. Meanwhile, the system is very compact and can be potentially packaged for portable applications. Since the use of soliton microcombs as ultralow noise microwave sources have drawn considerable interest in the communities of integrated photonics, the method described in this work should be appealing to a wide range of audience. Therefore, I would strongly recommend publication of this manuscript in Nature Communications.

Below are a few comments.

1. It's surprising that 1-km optical fibre can be wrapped on a few-cm diameter cylinder. The author should provide more information about how it is done.
2. Figure 1b might be a bit misleading. In time domain, free-running soliton pulses should have the same pulse width with the fibre-stabilized ones (like Figure 8, both in sech^2 shape). The authors should use another way to visualize timing jitter (like Figure 8).
3. How does the length of delay fiber affect the noise of stabilized solitons? To my understanding, phase noise of comb lines at very low-offset frequencies couldn't be very efficiently measured due to limited fiber length (a weak point of delayed self-heterodyne technique). Please clarify.
4. The author said '...corresponding to 1540 nm and 1560 nm, respectively) are filtered by fibre-Bragg gratings (with 2-nm full-width at half-maximum bandwidth)...' (cf. Page 19). Given that the spacing between two comb-lines is ~ 0.15 -nm (22-GHz), will the 2-nm full-width fibre-Bragg grating works as a good filter for a single comb-line?
5. m and n are selected to be 1540nm and 1560nm in the experiment. Will the selections of m & n affect the performance (e.g. from the single comb-line power) of the stabilizer as $\delta[\tau \times (m-n) f_{\text{rep}}]$ changes? The author should state the reasons for these selections.
6. The soliton is operated at quiet point to suppress noise transfer from pump detuning. However, in this case, is it essential since the repetition phase noise is detected and stabilized externally?
7. The author should discuss the noise sources that limit the performance of the stabilized soliton. Since the soliton is already at quiet point, does it suffer mostly from quantum noise [C. Bao, M.-G. Suh, B. Shen, K. Şafak, A. Dai, H. Wang, L. Wu, Z. Yuan, Q.-F. Yang, A. B. Matsko, F. X. Kärtner & K. Vahala "Quantum diffusion of microcavity solitons" Nature Physics (2021)] or intermode thermal noise [Yang, QF., Ji, QX., Wu, L. et al. Dispersive-wave induced noise limits in miniature soliton microwave sources. Nat Commun 12, 1442 (2021).] ?
8. The methods section reads a bit lengthy, and needs tighten up. Since the detection and stabilization system both use similar delayed self-heterodyne technique, I would suggest authors combine them into one part.

Some typos:

In Figure 8, next to the FBG it should be $v_n = n f_{rep} + f_{CEO}$.

Qi-Fan Yang

Reviewer #2:

Remarks to the Author:

The manuscript by Kwon et.al. describes the generation of stable microwaves using frequency comb generated in a mm-size optical microresonator.

The manuscript is generally well written and certainly of interest to the photonics community. However, I do not think that the work is novel enough and represents a significant advance in order to warrant publication in Nature Communications.

Below are more detailed comments:

1) This work is an extension from ref.19. In ref. 19, the authors used a similar fiber-delay-line based timing reference to evaluate the phase noise of the repetition rate of soliton microcombs generated in silica disk resonators. In ref.19, they locked the time delay of the fiber link to the soliton repetition rate with a very loose feedback bandwidth (300 Hz) and measured the phase noise outside of the locking bandwidth. In this work, instead, the authors lock the repetition rate of soliton microcomb to the fiber-delay-line based timing reference with a locking bandwidth of ~ 80 kHz. In my opinion, this work is not much of a technical advance over ref.19.

2) This work does not show better performance than other photonics-based microwave generation methods with compact packaging potential, commercial compact electro-optical frequency division-based oscillators (see e.g. <http://hqphotonics.com/>), MgF2 resonator-based microwave generation in ref.53, microwave generation based on free-running monolithic femtosecond laser (Opt. Express 25400-25409). Ref.53 generated a microwave signal with similar phase noise using a free-running, MgF2 resonator-based soliton microcomb, compared with that using a stabilized microcomb presented here. Commercial compact electro-optical frequency division-based oscillators (see <http://hqphotonics.com/>) have much better performance (-135 dBc/Hz at 1kHz).

3) Compared with the phase noise of electronic oscillators (Oven-controlled SAW oscillator-LNO10000B3, Rakon, SBO Compact Sapphire Oscillator), the method demonstrated here doesn't show an advancement either. In fact, the SBO Compact Sapphire Oscillator operating at room temperature can already provide much better phase noise than these presented here.

Some more technical comments are listed below:

4) since the system demonstrated here is still bulky setup, not on-chip, fully packaged, "ultracompact" is overclaimed. I also suggest the author rephrase the claim "ultralow-phase-noise".

5) on page 4, what is the bandwidth of the RF bandpass filter?

6) from my understanding, the AOFS1 is used not only for power kicking to generate soliton in the silica resonator, but also for stabilization of soliton repetition rate. Is this correct? If yes, how is it implemented?

7) In Fig.5, after power kicking, the pump-cavity detuning is locked via the laser PZT using soliton power as the feedback signal. Meanwhile, the soliton repetition rate is stabilized by changing pump-cavity detuning via frequency control of VCO applied to AOFS1. Do these two loops have cross talk between each other?

8) what is the reason that the RIN measurement around 20 kHz in Fig. 2b has a big jump?

9) on page 8, "To the best of our knowledge, this result corresponds to the lowest timing jitter and repetition-rate phase noise in the K-band generated from a free-running microcomb." Similar results have already been reported by the authors in ref.19.

10) in Fig.3c, both free-running and stabilized Allen deviations diverge after 0.4 s. What is the noise for time-dependent drift?

11) in Fig.4, please include the results reported in ref.53.

12) the authors used a Mach-Zehnder interferometer for timing jitter characterization, but a Michelson interferometer for stabilization of the soliton repetition rate. Please explain.

13) the authors discussed the potential of on-chip integration. However, the implementation of the FRMs on chip is a challenge, due to the magnet used inside.

14) on page 22, line 465, should be "two MUTC- photodiodes"

Reviewer #3:

Remarks to the Author:

This paper presents new and very interesting results on a stabilized micro-comb. The performance is impressive and the stabilization system is original (but already published and used on a more conventional fiber comb in 2015 by the same team). This paper must be accepted for publication.

I only have a few questions, mainly on the system performances.

1) In a stabilized system like this one, the limit in performance is the one of the frequency noise measurement system (if all the loop parameters could have been optimized). I am surprised to see in Figure 3 only white frequency noise close to the carrier on the stabilized microcomb phase noise spectrum (20 dB/dec, blue curve (ii)). What about the RIN conversion at the photodiodes level? Is there any $1/f$ noise in the RIN spectrum? (in figure 2, the RIN is only plotted above 1kHz offset). How the projected thermal noise curve (iii) in figure 3 is computed?

2) The stability performance plotted in figure 3c is also a bit surprising. You manage to get less than 10-12 relative Allan deviation on one second while your reference system (fiber frequency discriminator) is not thermally stabilized. Fiber is known to have poor performance in terms of relative index fluctuations versus temperature (in the range of 10^{-6} per Kelvin). I understand that one second is still short in time and may be the natural thermal isolation of the box in which the fiber is embedded is sufficient. Is there any passive temperature stabilization approach in this box? (such as an heavy piece of metal or anything with a large thermal constant).

3) In figure 5, I do not understand exactly how the locking on the soliton mode is performed. You say there is a detection of the power, but a detection of a power level does not give an error function centered on zero. Can you give more details on this point?

4) Line 169-171, you say that the result obtained on the free running microcomb is the best in terms of phase noise published in the K-band. It seems to be true for the type of resonator you are using. However, some results obtained with 3D whispering gallery mode resonators are better close to the carrier, even if they are not exactly at K band but at X band or Ka band. See the work of the OEwaves team: Liang et al., Nature communications, Aug. 2015, "High spectral purity Kerr...", or eventually a conference paper from D. Seidel et al. at MWP 2018 for results at 35 GHz. Of course,

the Q factor of a polished 3D resonator is very good but it is also very difficult to couple to an optical guide.

Point-by-point response to reviewers' comments

Reviewer 1

The manuscript reports a method for stabilizing repetition rate of soliton microcombs using a compact fiber-based timing reference. Such method allows detection and stabilization of the relative phase noise between two distantly located comb lines, which features orders of magnitude better sensitivity than using the detected soliton beatnote as an indicator. As a result, ultrastable 22-GHz microwave signal is generated from a chip-based silica microresonator, with more than 30 dB reduction in phase noise at low offset frequencies compared with free-running case. As a remark, the authors also compare two independent stabilized soliton microcombs to accurately reveal the improved long-term stability.

This work appears to be of high quality and has done adequate work to demonstrate the performance of this approach. The elaboration of key principles is straightforward and is well supported by detailed experiments and convincing analysis. Meanwhile, the system is very compact and can be potentially packaged for portable applications. Since the use of soliton microcombs as ultralow noise microwave sources have drawn considerable interest in the communities of integrated photonics, the method described in this work should be appealing to a wide range of audience. Therefore, I would strongly recommend publication of this manuscript in Nature Communications.

We thank Reviewer 1 for recognizing the importance of our work and supporting the publication of the paper.

1-1. It's surprising that 1-km optical fibre can be wrapped on a few-cm diameter cylinder. The author should provide more information about how it is done.

In the fiber-optic gyroscope community, kilometer-long fiber links have been wound with a few-cm diameter. For example, as described in one patent (US 8,773,665), a 5-km fiber could be wound with 80-mm diameter and 20-mm height. We leveraged this established fiber-winding method for making a compact fiber delayline in this work.

Revisions made: We added the following sentence in page 9 (lines 201-203): “The fibre link is wound with few-centimetre diameter (e.g., 7 cm diameter for 1-km-long fibre) using the fibre-winding method used for making fibre-optic gyroscopes.”

1-2. Figure 1b might be a bit misleading. In time domain, free-running soliton pulses should have the same pulse width with the fibre-stabilized ones (like Figure 8, both in sech² shape). The authors should use another way to visualize timing jitter (like Figure 8).

Revisions made: We modified Figure 1b as suggested by the reviewer.

1-3. How does the length of delay fiber affect the noise of stabilized solitons? To my understanding, phase noise of comb lines at very low-offset frequencies couldn't be very efficiently measured due to limited fiber length (a weak point of delayed self-heterodyne technique). Please clarify.

Regarding the impact of delay fiber length, in principle, longer fiber delay enables higher stability by enhancing the detection sensitivity (where the phase detection sensitivity scales with the delay time τ) and by lowering the thermomechanical noise of the fiber link. In our system, the achievable stability is mainly limited by the phase detection sensitivity, and lower noise is possible by longer delay τ . However, while the stabilization performance scales with the delay time τ , the achievable detection bandwidth

scales with $1/\tau$, therefore longer delay reduces the achievable locking bandwidth. For these reasons, there is a trade-off between detection sensitivity and bandwidth when selecting the optimal fiber delay length and, we selected 1-km link (2-km round-trip) for stabilization. Of course, depending on the intended application and noise performance, one can use longer or shorter fibre delayline (please see the revised Fig. S8 that shows this delay-dependent scaling in noise level and bandwidth).

Regarding the noise performance at very low offset frequency (e.g., <1 Hz), it will be fundamentally limited by the thermomechanical noise (*Phys. Rev. A* **86**, 023817 (2012), ref. 53), which scales with $1/f^3$. In practice, the thermally-induced length fluctuation including thermal expansion (with a coefficient of $\sim 10^{-6}/K$, as is also mentioned by Reviewer 3 in Point 3-2) can cause linear frequency drift and this mostly limits the performance (i.e., slow temperature drift over longer time scale simply expands the fiber length). The diverging stability in Allan deviation (shown in Fig. 3c) is the result of this linear frequency drift. There are several ways to reduce this, for example, using low-thermal-expansion fibers (from Linden Photonics or OFS Fitel as examples).

Revisions made:

We added discussion on fiber length selection and long-term drift by fiber thermal expansion in page 9 (lines 196-201) and 14 (lines 307-311) of the revised manuscript, respectively.

1-4. The author said ‘...corresponding to 1540 nm and 1560 nm, respectively) are filtered by fibre-Bragg gratings (with 2-nm full-width at half-maximum bandwidth)...’ (cf. Page 19). Given that the spacing between two comb-lines is ~ 0.15 -nm (22-GHz), will the 2-nm full-width fibre-Bragg grating works as a good filter for a single comb-line?

While single comb-line filtering is seemingly an ideal case, in practice, there are several issues. In our scheme, we need at least a few mW for effective self-heterodyning with sufficient SNR, therefore, EDFA is always required after the bandpass filters. If we filter a single comb-line, the signal power is too low (a few μ W) to achieve this level with sufficient SNR due to the amplified spontaneous emission (ASE) noise of the EDFA. Therefore, we need broader bandwidth filters for higher input power to the EDFA. Even in this case, because each comb-line can interfere with $2f_{AO}$ -shifted comb-line, this multiple-comb-line interference can contribute to higher SNR. Interestingly, when we tested 2 nm, 5 nm and 10 nm filters, the achieved SNR and resulting stabilized phase noise were almost identical.

Revisions made: We added the explanation on the selection of filter bandwidth in pages 3-4 of Supplementary Information.

1-5. v_m and v_n are selected to be 1540nm and 1560nm in the experiment. Will the selections of m & n affect the performance (e.g. from the single comb-line power) of the stabilizer as $\delta[\tau \times (m-n)f_{rep}]$ changes? The author should state the reasons for these selections.

As shown in the error signal formula, $\delta[\tau \times (m-n)f_{rep}]$, the phase detection sensitivity indeed scales with both the delay time τ and the frequency separation $(m-n)f_{rep} = v_m - v_n$. Therefore, increasing the delay τ and the frequency separation $(v_m - v_n)$ will increase the detection sensitivity and lower the phase noise. As shown in the response to Point 1-3, the impact of longer delay τ is already discussed.

It is clear that larger $(m-n)$ will enable higher detection sensitivity. However, if the frequency separation is too large beyond the optical bandwidth of soliton spectrum, the filtered power as well as SNR will decrease. As shown in Fig. S4, the combination of 1540 nm and 1560 nm provides a reasonable performance with less than 11 dB attenuation from the peak. When we use larger separation (e.g., 1530

nm and 1570 nm), the usable power is too low and lowers the SNR of the beating signal. In the future, it might be possible to use larger wavelength separation by broadening soliton spectrum and optimizing the EDFA (for example, using combination of C-band and L-band EDFAs) to cover broader optical spectrum.

Revisions made: We added Fig. S4 and added explanation on the selection of 1540 nm and 1560 nm in page 4 of Supplementary Information.

1-6. The soliton is operated at quiet point to suppress noise transfer from pump detuning. However, in this case, is it essential since the repetition phase noise is detected and stabilized externally?

There are two main reasons for operating the micro-comb near the quiet point. First, the phase noise above the locking bandwidth (in our work, ~120 kHz 3-dB bandwidth for 1-km delayline), the phase noise follows that of the free-running oscillator. Therefore, operating the comb near quiet point is necessary for achieving lower phase noise in the high offset frequency range. It is especially useful when using a longer fiber delayline for more phase noise suppression in the low offset frequency because longer delayline has lower locking bandwidth.

Second, even inside the locking bandwidth, the phase noise of stabilized microcomb is the phase noise of free-running comb divided by the loop gain of the feedback control system. The amount of achievable loop gain is typically limited mainly due to the limited phase margin. Thus, if we start from a worse free-running comb condition, the resulting stabilized phase noise will be also worse.

As a note, one interesting observation is that we could get the best stabilization result “near” the quiet point (e.g., ~500 kHz blue-detuned from the quiet point) and not at the exact quiet point. While the phase noise of free-running comb was indeed the lowest at the quiet point, interestingly, when we gradually detuned the microcomb from the quiet point, we found that the lowest phase noise of the stabilized comb is achieved at ~500 kHz blue-detuned point from the quiet point. It seems that, at the quiet point, the comb is the least sensitive to the pump disturbance and has a difficulty in increasing the loop gain when stabilizing. We believe that this finding will require more systematic study in the future.

Revisions made: We added discussion on the selection of comb operation near the quiet point in pages 8-9 (lines 183-190) of the revised manuscript.

1-7. The author should discuss the noise sources that limit the performance of the stabilized soliton. Since the soliton is already at quiet point, does it suffer mostly from quantum noise [C. Bao, M.-G. Suh, B. Shen, K. Şafak, A. Dai, H. Wang, L. Wu, Z. Yuan, Q.-F. Yang, A. B. Matsko, F. X. Kärtner & K. Vahala "Quantum diffusion of microcavity solitons" Nature Physics (2021)] or intermode thermal noise [Yang, QF., Ji, QX., Wu, L. et al. Dispersive-wave induced noise limits in miniature soliton microwave sources. Nat Commun 12, 1442 (2021).]?

Timing jitter of free-running microcombs will be fundamentally limited by the quantum noise [*Opt. Express* **21**, 28862 (2013) & *Nat. Physics* **17**, 462 (2021)] and the intermode thermal noise (i.e., thermorefractive noise) [*Phys. Rev. A* **99**, 061801 (2019) & *Nat. Commun.* **12**, 1442 (2021)]. By using the comb parameters, we plotted the projected quantum noise (curve (vi)) and thermorefractive noise (curve (vii)) along with the measured free-running (curve (i)) and fiber-stabilized (curves (ii)-(iv)) comb phase noise in Fig. S8 of Supplementary Information.

Regarding free-running jitter, the high-frequency phase noise closely approaches the quantum noise within 10 dB from ~30 kHz offset frequency and is limited by photodetection white noise floor (curve (viii)) from 300 kHz offset frequency. In the lower offset frequency (<30 kHz), the phase noise is limited by more technical noise. Although it is still a preliminary stage and beyond the scope of this paper, we believe that the pump frequency noise-originated jitter, which is coupled by the microresonator dispersion, is the dominant noise source (curve (v)). There will be a separate paper on this topic in the near future.

Regarding the stabilized jitter, which is the main question of the reviewer, the free-running jitter (which is limited by quantum noise) is the limiting factor for high frequency beyond the stabilization bandwidth (e.g. ~100 kHz for 1-km fiber case). Inside the stabilization bandwidth, as already discussed in Response 1-3, it is mostly limited by the fiber delay itself, not limited by the fundamental noise (such as quantum noise or thermorefractive noise) of the free-running combs. For the current system, it is mostly the phase detection sensitivity, which scales with delay length, and locking to longer fiber delay enables lower noise (as shown by curves (ii)-(iv) of Fig. S8, where the fibre delayline length is increased from 200-m through 1-km to 3-km).

Revisions made: We added Fig. S8 and added the explanation on the limiting factors of the free-running and stabilized jitter in pages 7-9 of the Supplementary Information.

1-8. The methods section reads a bit lengthy, and needs tighten up. Since the detection and stabilization system both use similar delayed self-heterodyne technique, I would suggest authors combine them into one part.

Revisions made: Following the reviewer's suggestion, we moved a large portion of *Methods* section to a separate Supplementary Information file and tighten up the contents of *Methods* section. We also combined the contents and figures for the detection and stabilization systems in Supplementary Information (for example, combining previous Figures 7 and 8 into a new Supplementary Figure S10).

1-9. In Figure 8, next to the FBG it should be $v_n = nf_{rep} + f_{CEO}$.

Revisions made: We fixed this typo (Figure S10 of Supplementary Information in the revised version).

Reviewer 2

The manuscript by Kwon et.al. describes the generation of stable microwaves using frequency comb generated in a mm-size optical microresonator.

*The manuscript is generally well written and certainly of interest to the photonics community. However, I do not think that the work is novel enough and represents a significant advance in order to warrant publication in *Nature Communications*.*

We thank the reviewer for detailed comments and feedback, and the quality of our manuscript is greatly improved by addressing them. However, we respectfully disagree with the reviewer's opinion on the novelty of our work.

While the stabilization of micro-combs is an important technology necessary for a variety of precision engineering fields, to our knowledge, very few works have tackled this problem. In particular, achieving high stability and low noise using a compact and mechanically robust setup is very important for more widespread applications of microcombs outside laboratory environment. In this work, we addressed this problem and could show compelling methods that can advance micro-combs in general and ultrahigh-Q micro-combs in particular.

The major novelty of our work can be summarized as below.

- We showed a new method that can tune and lock the repetition rate of ultrahigh-Q resonator comb (e.g., silica micro-comb) with broad bandwidth. So far, due to the lack of effective actuation mechanism and the necessity for continuous soliton-mode locking control, the control bandwidth was only <10 kHz for silica micro-combs. We combined a specially designed VCO and AOFS for achieving broadband repetition-rate tuning with >400 kHz bandwidth, which is at least ten times broader than previous works. Therefore, it opened up a way for achieving ultralow noise over the entire offset frequency range by combining ultralow timing jitter of ultrahigh-Q micro-combs and high phase stability of external reference sources in the high and low offset frequency ranges, respectively.
- We also demonstrated a compelling way for achieving both very low phase noise and compact platform using fiber photonics for stabilizing micro-combs. Compared to previous works based on ultrastable cavity-locked pump lasers or microwave frequency synthesizers, we could achieve excellent phase noise and phase stability from a compact platform (e.g., palm-sized footprint). We believe that this work showed a way to replace bulky or sophisticated reference sources to a more compact and lightweight device for more widespread applications of microcombs outside laboratory environment.
- The unique features of our method, generating highly stable and high-frequency optical pulse trains and microwaves from a compact device, is different from other low-noise photonic microwave generation methods, and can greatly improve or even create new application fields. As a compelling example, photonics-based radars, that combine optical pulse-based sampling/ADC and low-noise microwave generation, can be greatly benefited by having both high repetition-rate (>20 GHz) low-jitter optical pulse train and very low phase noise K-band microwave signals generated from a compact platform.

We remain confident that our work represents a major breakthrough in micro-comb and microwave photonics fields, and is highly suitable for publication in *Nature Communications*.

2-1. *This work is an extension from ref. 19. In ref. 19, the authors used a similar fiber-delay-line based timing reference to evaluate the phase noise of the repetition rate of soliton microcombs generated in silica disk resonators. In ref. 19, they locked the time delay of the fiber link to the soliton repetition rate with a very loose feedback bandwidth (300 Hz) and measured the phase noise outside of the locking bandwidth. In this work, instead, the authors lock the repetition rate of soliton microcomb to the fiber-delay-line based timing reference with a locking bandwidth of ~80 kHz. In my opinion, this work is not much of a technical advance over ref. 19.*

In fact, broadband locking of an ultrahigh-Q microcomb to an ultrastable reference is a highly nontrivial task and there were only few previous works that achieved locking of an ultrahigh-Q resonator (e.g., silica resonator). In ultrahigh-Q microcombs, it is necessary to stabilize the soliton mode from thermal destabilization using the active capture technique [*Optica* **2**, 1078 (2015) & *Nat. Photonics* **8**, 145 (2014)] or Pound-Drever-Hall (PDH) lock [*Phys. Rev. Lett.* **121**, 063902 (2018)]. Since high-bandwidth feedback control is already forced, it is challenging to suppress repetition-rate phase noise over broad bandwidth, mostly limited to only a few kHz so far. Very recently (in 2021), there was a report showing that the phase noise of silica microcomb can be stabilized to the microwave reference with ~10 kHz bandwidth [*Nat. Commun.* **12**, 1442 (2021)]. However, the phase noise above 10 kHz Fourier frequency is non-negligible in high-precision applications, and it is highly desirable to increase the locking bandwidth.

In this work, by combining a customized voltage-controlled oscillator (VCO) and an acousto-optic frequency shifter (AOFS) as a new actuator for controlling the repetition-rate of micro-comb, we could modulate both the frequency and amplitude of pump laser with high bandwidth. It could achieve an effective repetition-rate actuation with >400 kHz bandwidth. By using this new VCO+AOFS method, we could stabilize the micro-comb with ~120 kHz 3-dB locking bandwidth when using a 1-km fiber link (note that 80-kHz was the resonant peak position and we fixed this value to a more widely used 3-dB bandwidth in this revision), where the locking bandwidth is limited by the delay time of the fiber link itself, not by the actuator bandwidth.

By combining a new broadband repetition-rate tuning mechanism and a compact and stable fiber-based reference source, we could achieve low phase noise microwave and soliton-pulse generation from a silica micro-combs, which, we believe, is a major breakthrough in micro-combs and microwave photonics fields. Also note that this new repetition-rate actuation method can be used for other types of references (e.g., ultra-stable cavity or ultralow-noise microwave generator) and ultrahigh-Q micro-comb platforms (e.g., MgF₂ micro-combs) as well, and it can have a broader impact beyond a specific implementation of a stable microwave source.

Finally, compared to our previous ref. 19 (now ref. 24) work, while ref. 24 work used a PZT-stretcher as an actuator for fiber length tuning, which is straightforward and technically easy, the broadband repetition-rate tuning actuation of the ultrahigh-Q comb shown in this work represents a major advancement in micro-comb field, which was not shown in ref. 24 work.

Revisions made: We added more technical details on how to achieve broadband repetition-rate tuning for ultrahigh-Q resonator-based combs in Methods (lines 379-387) of the revised manuscript and pages 4-5 of the Supplementary Information. We also clarified the technical advances and meaning of such broadband repetition-rate tuning mechanism in pages 4 (lines 83-87) and 10-11 (lines 226-233) of the revised manuscript.

2-2. *This work does not show better performance than other photonics-based microwave generation methods with compact packaging potential, commercial compact electro-optical frequency division-*

based oscillators (see e.g. <http://hqphotonics.com/>), MgF₂ resonator-based microwave generation in ref.53, microwave generation based on free-running monolithic femtosecond laser (*Opt. Express* 25400-25409). Ref.53 generated a microwave signal with similar phase noise using a free-running, MgF₂ resonator-based soliton microcomb, compared with that using a stabilized microcomb presented here. Commercial compact electro-optical frequency division-based oscillators (see <http://hqphotonics.com/>) have much better performance (-135dBc/Hz at 1kHz).

We acknowledge that there are other compelling photonic technologies that can achieve similar or better microwave phase noise compared to our work, and we accordingly introduced these methods and properly cited these works in the revised manuscript.

As different methods and technologies have different characteristics and strengths, we feel that comparing the microwave phase noise values is not the only meaningful metric for assessing the importance of our work. Commercial electro-optic frequency division from hQphotonics shows really excellent microwave phase noise, however, it cannot generate low-jitter optical pulse train and the scalability in size seems to be rather limited from the current one, inferring from their *IEEE EFTF/IFCS 2017* paper (ref. 5). Monolithic mode-locked laser (ref. 6) indeed showed truly outstanding phase noise performance by exploiting solid-state mode-locked laser's intrinsically low quantum-limited jitter and removing low-frequency technical noise by CaF₂ spacer. However, the pulse repetition rate is rather low (~1 GHz) and generally requires higher pump power than silica micro-combs. Bulk MgF₂ resonator-based comb (refs. 32, 33, 59) is functionally similar to the silica comb used in this work and, as shown in Fig. 4, has a similar phase noise level with our free-running silica combs. Thus, as also discussed in response to Point 2-1, we believe that MgF₂ resonator can be also benefited from our work that showed both the broadband repetition-rate tuning mechanism for ultrahigh-Q resonators and the compact fiber-based reference source.

The unique features of our method, generating highly stable and high-frequency optical pulse trains and microwaves from a compact device, is different from other low-noise photonic microwave generation methods, and can greatly improve or even create new application fields. As a compelling example, photonics-based radars [Ghelfi et al, *Nature* **507**, 341 (2014); ref. 41], that combine optical pulse-based sampling/ADC and low-noise microwave generation, can be greatly benefited by having both high repetition-rate (>20 GHz) low-jitter optical pulse train and very low phase noise K-band microwave signals generated from a compact platform.

As discussed earlier in the response letter, the main contribution of our work is that we solved a major bottleneck in micro-comb field, i.e., how to lock ultrahigh-Q resonator to a stable reference with broad bandwidth, and further, how to replace bulky references to a compact yet high-stability device. Our result showed a way to tune repetition-rate of ultrahigh-Q resonator with broad bandwidth (at least ten times broader than previous works) and locked the micro-comb to a compact fiber photonic stabilizer with a palm-sized footprint. We believe that our result could provide an important missing ingredient in micro-comb community and can be applied for a wide range of other micro-combs as well.

Revisions made:

- a. We specified ultralow noise photonic microwave generation methods (electro-optic frequency division and monolithic mode-locked laser) with proper references cited in the introduction (page 2 lines 39-40).
- b. We added the best free-running MgF₂ comb result (shown in ref. 32) along with the silicon nitride comb result in page 3 (lines 64-66).

- c. We added the phase noise data of the MgF₂ comb (ref 53; new ref. 59 of the revised manuscript) for comparison as curve (vi) in Fig. 4.
- d. To emphasize the unique feature of our method – generation of both stable optical pulses and microwaves – we added “soliton-pulse generation” in the title and added the generation of low-jitter optical pulse train in the abstract and introduction (page 2).
- e. We added a more specific application scenario of our method for photonics radars in the discussion section (page 16 lines 343-347).

2-3. Compared with the phase noise of electronic oscillators (Oven-controlled SAW oscillator-LNO10000B3, Rakon, SBO Compact Sapphire Oscillator), the method demonstrated here doesn't show an advancement either. In fact, the SBO Compact Sapphire Oscillator operating at room temperature can already provide much better phase noise than these presented here.

We understand the reviewer's point and also appreciate that state-of-the-art microwave oscillators can indeed achieve ultralow phase noise that rivals or even surpasses our phase noise result.

As the reviewer will also agree, ultralow phase noise performance from state-of-the-art SAW oscillators and sapphire oscillators is the result of long research and development over several decades. In contrast, the use of micro-combs and related microwave photonic technology is a new, emerging research topic and has a long path ahead for further technical refinements in the future. Moreover, stabilized micro-combs can open up and improve new application fields, where traditional low-noise electronic oscillators cannot be used, for example, photonics-based radars, as already explained in the response to Comment 2-2.

2-4. since the system demonstrated here is still bulky setup, not on-chip, fully packaged, “ultracompact” is overclaimed. I also suggest the author rephrase the claim “ultralow-phase-noise”.

Revisions made: Following the reviewer's comment, we removed “ultracompact” from the title and the main text. We also modified “ultralow phase noise” to “low phase noise” in the abstract and main text.

2-5. on page 4, what is the bandwidth of the RF bandpass filter?

The bandwidth of the RF bandpass filter is 250 MHz.

Revisions made: We specified this number in the figure caption of Figure 1.

2-6. From my understanding, the AOFS1 is used not only for power kicking to generate soliton in the silica resonator, but also for stabilization of soliton repetition rate. Is this correct? If yes, how is it implemented?

As the reviewer rightly pointed out, AOFS₁ in Fig. 1 is used for both power kicking and stabilization. To implement both functions using AOFS₁, we customized the voltage-controlled oscillator (VCO) to modulate the power and frequency. The power kicking is conducted through the power modulation port on VCO, and the stabilization of the soliton repetition rate is conducted through the frequency modulation port on VCO (please see Fig. S5 of Supplementary Information).

As the frequency of the pump laser is already locked to the specific soliton mode via PZT, we applied the frequency modulation of VCO for the repetition-rate stabilization. Note that, previously, as demonstrated by Stone, J. *et al*, *Phys. Rev. Lett.* **121**, 063902 (2018), the repetition rate was stabilized

by modulating the intensity using an intensity modulator, but the achievable feedback bandwidth was only a few kHz. Therefore, we implemented the stabilization based on the frequency modulation for broadband feedback control loop.

Revisions made: We modified Figure 1a and Figure S1 (previous Fig. 5) and also added Figure S5 to clearly show how both the repetition-rate control and power modulation can be performed using the same AOFS. We also added more detailed information in pages 17-18 (in subsection “High-speed repetition-rate tuning mechanism”) of the revised manuscript.

2-7. In Fig.5, after power kicking, the pump-cavity detuning is locked via the laser PZT using soliton power as the feedback signal. Meanwhile, the soliton repetition rate is stabilized by changing pump-cavity detuning via frequency control of VCO applied to AOFS1. Do these two loops have cross talk between each other?

As the reviewer rightly pointed out, cross-talk between the two loops can be problematic for keeping two control loops simultaneously. Therefore, we quantified how much the repetition-rate stabilization control loop affects the maintenance of soliton mode-locking.

In order to quantify the conversion coefficient of pump-frequency to pump-intensity, we modulated the pump frequency by applying a 2-kHz sinewave to the VCO (see Fig. S6 of Supplementary Information). Then, we measured the pump frequency noise PSD (as shown in Opt. Lett. **34**, 914 (2009)) and the relative intensity noise (RIN) PSD of the comb output simultaneously. By comparing the 2-kHz modulation peaks of the pump frequency noise PSD and the comb RIN PSD, the pump frequency noise-to-comb RIN conversion coefficient (α) is obtained as 10^{-20} [1/Hz²].

Then, we measured the voltage noise PSD of the input signal to the VCO when the repetition-rate control loop is closed. By dividing this measured PSD by the sensitivity of the used VCO (0.8 MHz/V), we can convert the measured PSD to the pump frequency noise PSD of the stabilized repetition-rate control loop (see Fig. S7b of Supplementary Information). Finally, by using the pump frequency noise-to-comb RIN conversion coefficient ($\alpha = 10^{-20}$ [1/Hz²]), we can convert the pump frequency noise PSD into the equivalent comb-RIN PSD. As shown in Fig. S7c of Supplementary Information, the equivalent RIN induced by the repetition-rate control is >20 dB lower than the measured comb-RIN, which shows that the repetition rate stabilization does not hamper the soliton stabilization, and both control loops can operate simultaneously.

Revisions made: We stated that the crosstalk between two control loops is negligible in page 11 (lines 233-236) of the revised manuscript. We added detailed discussion (subsection “Crosstalk between soliton mode-locking and repetition-rate stabilization”) in pages 5-7 with Figures S6 and S7 in Supplementary Information.

2-8. What is the reason that the RIN measurement around 20 kHz in Fig. 2b has a big jump?

Due to the response of the PZT in the pump laser, the frequency noise of the pump laser has sudden jump around 20 kHz. In a high-Q microresonator, the frequency noise around 20 kHz is coupled into the intensity fluctuation.

Revisions made: We added this information in the caption of Figure 2b.

2-9. On page 8, “To the best of our knowledge, this result corresponds to the lowest timing jitter and repetition-rate phase noise in the K-band generated from a free-running microcomb.” Similar results have already been reported by the authors in ref. 19.

Following the reviewer’s comment, we removed that sentence. As a note, when comparing the free-running noise, the phase noise and integrated timing jitter (10 Hz – 1 MHz) are indeed 15 dB (at 75-kHz offset frequency) and a factor of 6 lower than ref. 19 (now ref. 24) result, respectively.

Revisions made: We removed the sentence.

2-10. In Fig.3c, both free-running and stabilized Allen deviations diverge after 0.4 s. What is the noise for time-dependent drift?

For the free-running microcomb, we believe that the time-dependent timing drift at this time scale is due to the thermal expansion of the silica material, which was also observed in silica-based cavity mirrors (D. Swierad et al, Sci. Rep. **6**, 33973 (2016)). For the fiber-stabilized microcomb, the stability of the repetition rate follows the timing stability of the fiber link. The SMF-28 fiber has a thermal expansion coefficient of $\sim 10^{-6}/\text{K}$ with a thermal coefficient of delay of $\sim 40\text{-}130\text{ ps/km/K}$ (as is also discussed in the response for Reviewer 3’s Comment 3-2), and this time-dependent temperature drift in the fiber link induces Allan deviation diverging after $\sim 0.4\text{ s}$.

Revisions made: We added the discussion in page 14 (lines 305-311) of the revised manuscript.

2-11. in Fig.4, please include the results reported in ref.53.

Revisions made: We include the phase noise PSD of the MgF_2 microcomb (ref. 59 of revised manuscript; ref. 53 of previous manuscript) as curve (vi) in new Fig. 4.

2-12. the authors used a Mach-Zehnder interferometer for timing jitter characterization, but a Michelson interferometer for stabilization of the soliton repetition rate. Please explain.

The function of both interferometers is exactly same and interchangeable. In this work, the reason why we used the Mach-Zehnder interferometer for free-running jitter measurement was to obtain twice more measurement bandwidth than Michelson interferometer when using the same fiber delay-line spool. In this case, although the measurement sensitivity becomes twice worse, it did not limit the measurement of free-running jitter, so we could measure the jitter over broader offset frequency range.

Revisions made: We added the explanation on this part in page 12 of Supplementary Information.

2-13. the authors discussed the potential of on-chip integration. However, the implementation of the FRMs on chip is a challenge, due to the magnet used inside.

Indeed, the implementation of the magnetic component such as FRM and acousto-optic frequency shifter (AOFS) on a chip is challenging. On one hand, FRM may not be necessary for on-chip integration because it will be generally a polarization-maintaining waveguide structure in the integrated chip. On the other hand, we can also implement the on-chip version using a Mach-Zehnder interferometer structure instead of a Michelson interferometer. Regarding the AOFS, it can be replaced by a single-

sideband phase modulator, which can be implemented as an on-chip device (as shown in Yu. B.-M. et al., *Photonics Research* **6**, 6 (2018), which is Ref. 48 of the revised manuscript).

2-14. on page 22, line 465, should be "two MUTC- photodiodes"

Revisions made: We modified "an MUTC-photodiode" to "two MUTC-photodiodes" (in page 13 of Supplementary Information).

Reviewer 3

This paper presents new and very interesting results on a stabilized micro-comb. The performance is impressive and the stabilization system is original (but already published and used on a more conventional fiber comb in 2015 by the same team). This paper must be accepted for publication. I only have a few questions, mainly on the system performances.

We thank Reviewer 3 for recognizing the importance of our work and supporting the publication of the paper.

3-1. In a stabilized system like this one, the limit in performance is the one of the frequency noise measurement system (if all the loop parameters could have been optimized). I am surprised to see in Figure 3 only white frequency noise close to the carrier on the stabilized microcomb phase noise spectrum (20 dB/dec, blue curve (ii)). What about the RIN conversion at the photodiodes level? Is there any 1/f noise in the RIN spectrum? (in figure 2, the RIN is only plotted above 1kHz offset). How the projected thermal noise curve (iii) in figure 3 is computed?

Regarding white frequency noise at low offset frequency: As the reviewer pointed out, the stabilized phase noise shows white frequency noise-limited performance (i.e., -20 dB/dec phase noise slope) down to 10 Hz offset frequency. It is because the current performance is limited by the phase detection sensitivity of the fiber stabilizer (as discussed in Responses to 1-3 and 1-7 comments), and the equivalent white noise floor of the interference signal is higher than the fundamental thermal noise of fiber delay-line and technical noise (such as vibration noise) coupled to the fiber delay-line reference itself. As will be shown below, the photodetection noise is also lower than the measured performance.

Regarding RIN conversion of photodiodes: To assess the impact of RIN-converted phase noise, we measured the amplitude-to-phase conversion coefficient (APC) of the used photodiode by modulating the comb intensity and measuring the 22-GHz phase noise after photodetection. At 10, 100 and 200 kHz Fourier frequency, the APC was measured to be 8 rad/%. From the measured APC, the projected phase noise due to comb RIN is plotted in Fig. S9 of Supplementary Information. The RIN-converted photodetection noise is below the stabilized performance and does not limit the measurement.

Regarding 1/f noise of RIN: As shown in the figure below, there is no 1/f noise from 10 Hz to 1 kHz in the RIN spectrum as well.

Figure R1

Regarding the projected thermal noise curve (iii): The projected thermal noise (curve (iii) in Fig. 3a) includes the thermomechanical length fluctuation [*Electron. Lett.* **46**, 1515 (2010)] and thermoconductive length fluctuation [*Phys. Rev. A* **86**, 023817 (2012), ref. 51]. We compute the thermal noise-induced phase error in the 2 km-long (due to round trip) fiber interferometer ($S_{\phi_{thermal}}(f)$). This

phase error is converted to the laser frequency noise as $S_v(f) = \left(\frac{1}{\tau}\right)^2 S_{\phi_{thermal}}(f)$, where τ is the timing delay induced by the interferometer. In this experiment, the common-mode f_{ceo} noise is rejected so that the laser frequency noise ($S_v(f)$) corresponds to 2.5 THz carrier (i.e., $(m-n)f_{rep}$). Finally, the frequency noise of 2.5 THz is converted to the phase noise of 22 GHz.

Revisions made:

- a. Regarding the noise in the low offset frequency and the scalability, we added discussion in page 13 (lines 282-286) of the revised manuscript and pages 8-9 of the Supplementary Information.
- b. Regarding RIN conversion of photodiode and its impact, we added Figure S9 and discussion in page 9 of the Supplementary Information.
- c. Regarding the projected thermal noise curve, we added more information in page 9-10 of the Supplementary Information.

3-2. The stability performance plotted in figure 3c is also a bit surprising. You manage to get less than 10-12 relative Allan deviation on one second while your reference system (fiber frequency discriminator) is not thermally stabilized. Fiber is known to have poor performance in terms of relative index fluctuations versus temperature (in the range of 10⁻⁶ per Kelvin). I understand that one second is still short in time and may be the natural thermal isolation of the box in which the fiber is embedded is sufficient. Is there any passive temperature stabilization approach in this box? (such as a heavy piece of metal or anything with a large thermal constant).

As the reviewer rightly pointed out, relatively large thermo-optic coefficient of fiber ($\sim 10^{-6} \text{ K}^{-1}$) can cause timing drift, which is also observed in the Allan deviation (Fig. 3c) after ~ 0.4 s averaging time. For shorter time scale, as also explained in Comment 3-1, the stabilized performance is limited by the white frequency noise of the interference signal, which is higher than the thermomechanical and thermoconductive noise.

To minimize the impact of acoustic noise and thermal drift coupled to the fiber delay-line, we packaged the fiber-stabilizer in an air-tight box to isolate it from the laboratory environment. Although we did not use a heavy piece of metal or special material with large thermal constant to keep the stabilization system compact, we found that this air-tight box works fairly well. Also note that we performed the measurement of noise and stability after waiting for several hours from turning on the system, so that the entire electronics, fiber and micro-comb systems reach a fairly thermal equilibrium state. Of course, for the future use outside laboratory environment, a proper temperature control of the fiber stabilizer box (using a TEC control, for example) will be required.

As an additional note, we improved the fiber stabilizer's packaging method and size from 130 mm x 90 mm x 65 mm to 111 mm x 85 mm x 56 mm in the revision, and replaced Fig. 1a inset photo to the new package.

Revisions made: We added more information in page 14 (lines 307-311) of the revised manuscript and in page 10 of Supplementary Information. We also replaced the photo of the fiber stabilizer in Fig. 1a to a new one and revised the package size.

3-3. In figure 5, I do not understand exactly how the locking on the soliton mode is performed. You say there is a detection of the power, but a detection of a power level does not give an error function centered on zero. Can you give more details on this point?

The comb power detected by the photodiode in Fig. 5 (new Supplementary Figure Fig. S1) changes when the laser frequency is swept over time. We set the locking point by the input offset voltage of the loop filter. The error signal in the loop filter corresponds to the difference between the set voltage (V_{ref}) and the comb power (V_{in}), that is, $V_{error} = V_{in} - V_{ref}$. If the comb power fluctuates, the laser frequency is modulated. The feedback is conducted to maintain V_{error} to zero.

Revisions made: We summarized this process in Fig. S2 and added explanation in page 1 of Supplementary Information.

3-4. Line 169-171, you say that the result obtained on the free running microcomb is the best in terms of phase noise published in the K-band. It seems to be true for the type of resonator you are using. However, some results obtained with 3D whispering gallery mode resonators are better close to the carrier, even if they are not exactly at K band but at X band or Ka band. See the work of the OEWaves team: Liang et al., Nature communications, Aug. 2015, "High spectral purity Kerr...", or eventually a conference paper from D. Seidel et al. at MWP 2018 for results at 35 GHz. Of course, the Q factor of a polished 3D resonator is very good but it is also very difficult to couple to an optical guide.

Revisions made: We removed the sentence "To the best of our knowledge, this result corresponds to the lowest timing jitter and repetition-rate phase noise in the K-band generated from a free-running microcomb." We added the phase noise of ref. 59 (Liang et al, Nature Commun 2015) in the revised Fig. 4 for comparison, following the Comment 2-11 of Reviewer 2. We also added Seidel et al, MWP 2018 paper as new ref. 36.

Reviewer #1 (Remarks to the Author):

After the last review, this manuscript appears to be smoothly written and well-organized. The authors have revealed all the pros and cons of their approach with comparative tests and convincing analysis. Due to the comprehensive experimental details displayed, this scheme will be quickly accepted and promote the development of microcomb stabilization. As the integrated ultrastable microwave source is of great importance, the stabilizing method described in this work should be appealing to both academia and industry. Therefore, I would recommend publishing this manuscript in Nature Communications.

Two unsettled details:

1. As is mentioned in the section of OE conversion and Ref. 52, the soliton pulsewidth is compressed to <1 ps using a dispersion compensating fibre to avoid a pulsewidth-dependent noise floor at high offset frequencies. But why the shot noise level of the 22-GHz microwave in this work has not reached the CW shot noise level $L(f) = eI_{ave}R_{load}/P_{PF} \simeq -160\text{dBc/Hz}$ (mentioned in Ref. 59, 50 Ohm RF load assumed). As is shown in the Ref. 59, the RF power extracted from the fast PD is $\sim -13\text{dBm}$ ($50\mu\text{W}$, $\sim 11\text{dB}$ weaker than here in this work) and is able to reach the $\sim -150\text{dBc/Hz}$ level shot noise at high offset frequencies (almost the same as this work).

Moreover, it has been shown that the 1st harmonic microwaves generated by Gaussian pulses with 1 ps pulsewidth can have $\sim -5\text{dBc/Hz}$ suppressed shot noise level than the CW shot noise level in previous literatures (e.g. Quinlan, F., Fortier, T., Jiang, H. et al. Exploiting shot noise correlations in the photodetection of ultrashort optical pulse trains. Nature Photon 7, 290–293 (2013). <https://doi.org/10.1038/nphoton.2013.33>). Please explain such discrepancy.

2. In the Fig.1, the electrical signal after two BPDs and AMPs can be expressed as

$$V_m(t) = \sin[2\omega_{AO}t + \phi_m(t + 2\tau) - \phi_m(t) + \phi_{0m}]$$

and

$$V_n(t) = \sin[2\omega_{AO}t + \phi_n(t + 2\tau) - \phi_n(t) + \phi_{0n}]$$

where ω_{AO} is the frequency shift of the AOFS, $\phi_{m,n}(t)$ are the time-domain phase noise of the m^{th} and n^{th} comb lines, and $\phi_{0m,n}$ are the constant phase biases of the electrical signals.

Then, the electrical signal that drives the VCO can be expressed as

$$V_{VCO}(t) = \sin[\phi_n(t + 2\tau) - \phi_n(t) - \phi_m(t + 2\tau) + \phi_m(t) + \phi_{0n} - \phi_{0m}]$$

Through adjusting the phase biases to nullify the DC error voltage, the feedback loop will lock the feedback signal to the residual noise of the loop. Then it follows

$$\phi_n(t + 2\tau) - \phi_n(t) - \phi_m(t + 2\tau) + \phi_m(t) = \Delta\phi(t + 2\tau) - \Delta\phi(t) \simeq \xi(t)$$

where $\Delta\phi(t) = \phi_n(t) - \phi_m(t)$, and $\xi(t)$ is the time-domain residual noise signal of the loop, taking $\sin(x) \simeq x$ ($x \ll 1$) into account. The stationary autocorrelation function $R(t)$ can be expressed as

$$R_\xi(t) = 2R_{\Delta\phi}(t) - R_{\Delta\phi}(t + 2\tau) - R_{\Delta\phi}(t - 2\tau)$$

Wiener–Khinchin theorem suggests the single sideband power spectrum density $S(f)$ to be

$$S_\xi(f) = 4 \sin^2(2\pi f\tau) S_{\Delta\phi}(f)$$

And the phase noise of the 22-GHz microwave can be expressed as

$$L(f) = \frac{S_{\Delta\phi}(f)}{2(m-n)^2} \simeq \frac{1}{32\pi^2(m-n)^2\tau^2} \times \frac{S_{\xi}(f)}{f^2} \sim \frac{S_{\xi}(f)}{f^2}$$

when considering the $f \ll 1/\tau$ cases. If $S_{\xi}(f) \simeq \text{const}$, the phase noise of the 22-GHz microwave will roll off following $1/f^2$ trend as is shown in the Fig. 2. Therefore, I would suspect the limitation of current system is set by the locking residuals.

Reviewer #2 (Remarks to the Author):

Thanks for the detailed response. The authors' main argument is that they achieve a higher locking bandwidth (100 kHz) by combining a voltage-controlled oscillator and an acousto-optic frequency shifter (AOFS), instead of the use of a PZT stretcher (10 kHz) in ref. 24.

This improvement is a very straightforward idea and merely a technical improvement. In fact, similar laser frequency control by using voltage-controlled oscillators and a frequency shifter has been described in Phys. Rev. Lett. 121, 063902, 2018. In addition, there are many techniques to control laser frequency at microsecond timescales.

In conclusion, this doesn't convince me that the revised manuscript has enough novelty. I think readers of the Nat. Commun. would expect more scientific advance, not just an improvement of locking bandwidth. This work is an extension from their previous work that is cited as Ref. 24.

Reviewer #3 (Remarks to the Author):

The authors have correctly answered my questions and added the supplementary information required to better understand these points. As I said in my first review, it is a new and very interesting work that must be accepted for publication in Nature Com.

Point-by-point response to reviewers' comments

Reviewer 1

After the last review, this manuscript appears to be smoothly written and well-organized. The authors have revealed all the pros and cons of their approach with comparative tests and convincing analysis. Due to the comprehensive experimental details displayed, this scheme will be quickly accepted and promote the development of microcomb stabilization. As the integrated ultrastable microwave source is of great importance, the stabilizing method described in this work should be appealing to both academia and industry. Therefore, I would recommend publishing this manuscript in Nature Communications.

We thank the reviewer for recognizing the importance of our work and supporting the publication.

1. As is mentioned in the section of OE conversion and Ref. 52, the soliton pulsewidth is compressed to <1 ps using a dispersion compensating fibre to avoid a pulsewidth-dependent noise floor at high offset frequencies. But why the shot noise level of the 22-GHz microwave in this work has not reached the CW shot noise level $L(f) = eI_{\text{ave}}R_{\text{load}}/P_{\text{PF}} \approx -160$ dBc/Hz (mentioned in Ref. 59, 50 Ohm RF load assumed). As is shown in the Ref. 59, the RF power extracted from the fast PD is ~ -13 dBm ($50 \mu\text{W}$, ~ 11 dBm weaker than here in this work) and is able to reach the ~ -150 dBc/Hz level shot noise at high offset frequencies (almost the same as this work). Moreover, it has been shown that the 1st harmonic microwaves generated by Gaussian pulses with 1 ps pulsewidth can have ~ -5 dBc/Hz suppressed shot noise level than the CW shot noise level in previous literatures (e.g. Quinlan, F., Fortier, T., Jiang, H. et al. Exploiting shot noise correlations in the photodetection of ultrashort optical pulse trains. Nature Photon 7, 290–293 (2013). <https://doi.org/10.1038/nphoton.2013.33>). Please explain such discrepancy.

The measured photodetection white noise floor is -153 dBc/Hz, which is ~ 8 dB higher than the calculated CW shot noise floor using the relationship, $L(f) = eI_{\text{ave}}R_{\text{load}}/P_{\text{PF}} = -161$ dBc/Hz, where I_{ave} is ~ 7 mA, R_{load} is 50 Ohm, and P_{PF} is ~ 1 mW (~ 0 dBm). Note that the thermal noise is calculated to be -170 dBc/Hz, which is ~ 9 dB lower than the shot noise, and the photodetection noise is mostly limited by the shot noise.

In our experiment, to obtain the phase noise and frequency instability data of the generated microwave (shown in Fig. 3), we used an RF amplifier after the photodetection. The used RF amplifier had a ~ 6 -dB noise figure at 22-GHz carrier, which mainly contributed to the noise floor degradation from the calculated shot noise floor. In addition to the noise figure, we also believe that the amplitude-to-phase conversion in the RF amplifier further exacerbates the phase noise, which eventually worsens the phase noise by ~ 8 dB.

As the reviewer pointed out, the shot noise floor can be improved by utilizing short pulse duration (as shown in Nat. Photonics 7, 290 (2013)). The bandwidth of the used photodetector (22 GHz) was, however, similar to the fundamental repetition rate of the microcomb. As a result, the shot noise suppression could not be fully utilized due to the limited bandwidth of the photodiode.

Revisions made: We added the following sentence in Lines 278-281 (page 13) of the main manuscript: “Note that, in the high Fourier frequency (>300 kHz), the measured photodetection noise floor (~ -153 dBc/Hz) is degraded from the projected shot noise limit (-161 dBc/Hz) by the noise figure as well as amplitude-to-phase conversion of the used RF amplifier.”

2. In the Fig.1, the electrical signal after two BPDs and AMPs can be expressed as

$$V_m(t) = \sin [2\omega_{AO}t + \phi_m(t + 2\tau) - \phi_m(t) + \phi_{om}]$$

and

$$V_n(t) = \sin [2\omega_{AO}t + \phi_n(t + 2\tau) - \phi_n(t) + \phi_{on}]$$

where ω_{AO} is the frequency shift of the AOFs, $\phi_{m,n}(t)$ are the the time-domain phase noise of the m th and n th comb lines, and $\phi_{0m,n}$ are the constant phase biases of the electrical signals.

Then, the electrical signal that drives the VCO can be expressed as

$$V_{VCO}(t) = \sin [\phi_n(t + 2\tau) - \phi_n(t) - \phi_m(t + 2\tau) + \phi_m(t) + \phi_{on} - \phi_{om}]$$

Through adjusting the phase biases to nullify the DC error voltage, the feedback loop will lock the feedback signal to the residual noise of the loop. Then it follows

$$\phi_n(t + 2\tau) - \phi_n(t) - \phi_m(t + 2\tau) + \phi_m(t) = \Delta\phi(t + 2\tau) - \Delta\phi(t) = \xi(t)$$

where $\Delta\phi(t) = \phi_n(t) - \phi_m(t)$, and $\xi(t)$ is the time-domain residual noise signal of the loop, taking $\sin(x) = x(x \ll 1)$ into account. The stationary autocorrelation function $R(t)$ can be expressed as

$$R_\xi(t) = 2R_{\Delta\phi}(t) - R_{\Delta\phi}(t + 2\tau) - R_{\Delta\phi}(t - 2\tau)$$

Wiener–Khinchin theorem suggests the single sideband power spectrum density $S(f)$ to be

$$S_\xi(f) = 4 \sin^2 2\pi f\tau S_{\Delta\phi}(f)$$

And the phase noise of the 22-GHz microwave can be expressed as

$$L(f) = \frac{S_{\Delta\phi}(f)}{2(m-n)^2} \approx \frac{1}{32\pi^2(m-n)^2\tau^2} \times \frac{S_\xi(f)}{f^2} \sim \frac{S_\xi(f)}{f^2}$$

when considering the $f \ll 1/\tau$ cases. If $S_\xi(f) \approx \text{const}$, the phase noise of the 22-GHz microwave will roll off following $1/f^2$ trend as is shown in the Fig. 2. Therefore, I would suspect the limitation of current system is set by the locking residuals.

We thank the reviewer for the time and dedication for detailed derivation of the residual phase noise. As the reviewer correctly derived, the locking residuals indeed set the limitation of the current system, following the $1/f^2$ trend as shown in Fig. 3a and Fig. S8 data.

The measured residual frequency noise has a white noise characteristic, leading to the $\frac{S_\xi(f)}{f^2}$ phase noise characteristic. Here, the residual frequency noise PSD is mostly limited by the Rayleigh scattering-limited relative intensity noise (RIN) of the fiber link at $2f_{AO}$, where the conversion of the Rayleigh scattering RIN to the frequency noise PSD scales with the fiber delay (τ) [Cahill et al, Opt. Express 23, 6400 (2015)]. On the other hand, the impact of timing detection sensitivity scaling to the frequency noise PSD inversely scales with the square of the fiber length ($1/\tau^2$) because the timing detection sensitivity of the fiber link scales with the fiber delay. As a result, when the Rayleigh scattering RIN scaling and timing detection sensitivity scaling are combined, the resulting frequency noise PSD inversely scales with the delay ($S_\xi(f) \sim 1/\tau$) as shown in Figure S8.

Revisions made: As the reason for the limited noise performance, we added “the Rayleigh scattering-limited intensity noise to the frequency noise conversion⁵⁴” in Lines 286-287 (page 13) with new ref. 54 [Cahill et al, Opt. Express 23, 6400 (2015)]. More detailed explanation outlined above is added in Supplementary Information (in page 9): “The main reason for the residual white noise floor (in frequency noise PSD) is the Rayleigh scattering RIN. While the Rayleigh scattering PSD scales with the delay length¹⁰, the enhanced detection sensitivity (which scales to the delay length) improves the phase noise PSD by the square of the delay length. As a result, when the Rayleigh scattering RIN scaling and timing detection sensitivity scaling are combined, the resulting frequency noise PSD floor inversely scales with the fibre delay length.”

Reviewer 2

Thanks for the detailed response. The authors' main argument is that they achieve a higher locking bandwidth (100 kHz) by combining a voltage-controlled oscillator and an acousto-optic frequency shifter (AOFS), instead of the use of a PZT stretcher (10 kHz) in ref. 24.

*This improvement is a very straightforward idea and merely a technical improvement. In fact, similar laser frequency control by using voltage-controlled oscillators and a frequency shifter has been described in *Phys. Rev. Lett.* 121, 063902, 2018. In addition, there are many techniques to control laser frequency at microsecond timescales.*

*In conclusion, this doesn't convince me that the revised manuscript has enough novelty. I think readers of the *Nat. Commun.* would expect more scientific advance, not just an improvement of locking bandwidth. This work is an extension from their previous work that is cited as Ref. 24.*

We respectfully disagree with the reviewer's opinion on the novelty of our work. The major contribution of our study is to provide a compelling approach for stabilizing the timing of ultrahigh-Q micro-combs on a simple, compact and robust platform, allowing chip-scale micro-combs to be used to their full potential. To our knowledge, very few studies have sought for such a compact yet high-performance micro-comb stabilization method, and we presented a novel approach with one of the lowest microwave phase noise performances for micro-comb systems. We remain confident that our result represents a major breakthrough in micro-comb and microwave photonics fields.

In addition to the overall impact and novelty aspects outlined above, we'd like to address the reviewer's comments on the frequency control methods. While the method shown in *Phys. Rev. Lett.* 121, 063902 (2018) appears to use a similar frequency control approach, a closer look reveals that it is different from our approach and, as a result, could not demonstrate broadband stabilization. There are many techniques for controlling the laser frequency at a microsecond timescale, as the reviewer pointed out, but there has been no work that accomplished broadband frequency modulation and stabilization for ultrahigh-Q microresonators. It has been particularly challenging because such ultrahigh-Q microresonators require continuous soliton-mode stabilization against thermal destabilization. As a result, broadband locking of repetition-rate in ultrahigh-Q microresonators has not been straightforward, and our result in this study represents one of the major advances in ultrahigh-Q microcombs compared to our previous work (ref. 24) and other laser frequency control technique studies.

Reviewer 3

*The authors have correctly answered my questions and added the supplementary information required to better understand these points. As I said in my first review, it is a new and very interesting work that must be accepted for publication in *Nature Com.**

We thank the reviewer for recognizing the importance of our work and supporting the publication.

REVIEWERS' COMMENTS

Reviewer #1 (Remarks to the Author):

The authors have satisfactorily addressed all my comments. The manuscript should be published as it is.

Point-by-point response to reviewers' comments

Reviewer 1

After the last review, this manuscript appears to be smoothly written and well-organized. The authors have revealed all the pros and cons of their approach with comparative tests and convincing analysis. Due to the comprehensive experimental details displayed, this scheme will be quickly accepted and promote the development of microcomb stabilization. As the integrated ultrastable microwave source is of great importance, the stabilizing method described in this work should be appealing to both academia and industry. Therefore, I would recommend publishing this manuscript in Nature Communications.

We thank the reviewer for recognizing the importance of our work and supporting the publication.

1. As is mentioned in the section of OE conversion and Ref. 52, the soliton pulsewidth is compressed to <1 ps using a dispersion compensating fibre to avoid a pulsewidth-dependent noise floor at high offset frequencies. But why the shot noise level of the 22-GHz microwave in this work has not reached the CW shot noise level $L(f) = eI_{\text{ave}}R_{\text{load}}/P_{\text{PF}} \approx -160$ dBc/Hz (mentioned in Ref. 59, 50 Ohm RF load assumed). As is shown in the Ref. 59, the RF power extracted from the fast PD is ~ -13 dBm ($50 \mu\text{W}$, ~ 11 dBm weaker than here in this work) and is able to reach the ~ -150 dBc/Hz level shot noise at high offset frequencies (almost the same as this work). Moreover, it has been shown that the 1st harmonic microwaves generated by Gaussian pulses with 1 ps pulsewidth can have ~ -5 dBc/Hz suppressed shot noise level than the CW shot noise level in previous literatures (e.g. Quinlan, F., Fortier, T., Jiang, H. et al. Exploiting shot noise correlations in the photodetection of ultrashort optical pulse trains. Nature Photon 7, 290–293 (2013). <https://doi.org/10.1038/nphoton.2013.33>). Please explain such discrepancy.

The measured photodetection white noise floor is -153 dBc/Hz, which is ~ 8 dB higher than the calculated CW shot noise floor using the relationship, $L(f) = eI_{\text{ave}}R_{\text{load}}/P_{\text{PF}} = -161$ dBc/Hz, where I_{ave} is ~ 7 mA, R_{load} is 50 Ohm, and P_{PF} is ~ 1 mW (~ 0 dBm). Note that the thermal noise is calculated to be -170 dBc/Hz, which is ~ 9 dB lower than the shot noise, and the photodetection noise is mostly limited by the shot noise.

In our experiment, to obtain the phase noise and frequency instability data of the generated microwave (shown in Fig. 3), we used an RF amplifier after the photodetection. The used RF amplifier had a ~ 6 -dB noise figure at 22-GHz carrier, which mainly contributed to the noise floor degradation from the calculated shot noise floor. In addition to the noise figure, we also believe that the amplitude-to-phase conversion in the RF amplifier further exacerbates the phase noise, which eventually worsens the phase noise by ~ 8 dB.

As the reviewer pointed out, the shot noise floor can be improved by utilizing short pulse duration (as shown in Nat. Photonics 7, 290 (2013)). The bandwidth of the used photodetector (22 GHz) was, however, similar to the fundamental repetition rate of the microcomb. As a result, the shot noise suppression could not be fully utilized due to the limited bandwidth of the photodiode.

Revisions made: We added the following sentence in Lines 278-281 (page 13) of the main manuscript: “Note that, in the high Fourier frequency (>300 kHz), the measured photodetection noise floor (~ -153 dBc/Hz) is degraded from the projected shot noise limit (-161 dBc/Hz) by the noise figure as well as amplitude-to-phase conversion of the used RF amplifier.”

2. In the Fig.1, the electrical signal after two BPDs and AMPs can be expressed as

$$V_m(t) = \sin [2\omega_{AO}t + \phi_m(t + 2\tau) - \phi_m(t) + \phi_{om}]$$

and

$$V_n(t) = \sin [2\omega_{AO}t + \phi_n(t + 2\tau) - \phi_n(t) + \phi_{on}]$$

where ω_{AO} is the frequency shift of the AOFs, $\phi_{m,n}(t)$ are the the time-domain phase noise of the m th and n th comb lines, and $\phi_{0m,n}$ are the constant phase biases of the electrical signals.

Then, the electrical signal that drives the VCO can be expressed as

$$V_{VCO}(t) = \sin [\phi_n(t + 2\tau) - \phi_n(t) - \phi_m(t + 2\tau) + \phi_m(t) + \phi_{on} - \phi_{om}]$$

Through adjusting the phase biases to nullify the DC error voltage, the feedback loop will lock the feedback signal to the residual noise of the loop. Then it follows

$$\phi_n(t + 2\tau) - \phi_n(t) - \phi_m(t + 2\tau) + \phi_m(t) = \Delta\phi(t + 2\tau) - \Delta\phi(t) = \xi(t)$$

where $\Delta\phi(t) = \phi_n(t) - \phi_m(t)$, and $\xi(t)$ is the time-domain residual noise signal of the loop, taking $\sin(x) = x(x \ll 1)$ into account. The stationary autocorrelation function $R(t)$ can be expressed as

$$R_\xi(t) = 2R_{\Delta\phi}(t) - R_{\Delta\phi}(t + 2\tau) - R_{\Delta\phi}(t - 2\tau)$$

Wiener–Khinchin theorem suggests the single sideband power spectrum density $S(f)$ to be

$$S_\xi(f) = 4 \sin^2 2\pi f\tau S_{\Delta\phi}(f)$$

And the phase noise of the 22-GHz microwave can be expressed as

$$L(f) = \frac{S_{\Delta\phi}(f)}{2(m-n)^2} \approx \frac{1}{32\pi^2(m-n)^2\tau^2} \times \frac{S_\xi(f)}{f^2} \sim \frac{S_\xi(f)}{f^2}$$

when considering the $f \ll 1/\tau$ cases. If $S_\xi(f) \approx \text{const}$, the phase noise of the 22-GHz microwave will roll off following $1/f^2$ trend as is shown in the Fig. 2. Therefore, I would suspect the limitation of current system is set by the locking residuals.

We thank the reviewer for the time and dedication for detailed derivation of the residual phase noise. As the reviewer correctly derived, the locking residuals indeed set the limitation of the current system, following the $1/f^2$ trend as shown in Fig. 3a and Fig. S8 data.

The measured residual frequency noise has a white noise characteristic, leading to the $\frac{S_\xi(f)}{f^2}$ phase noise characteristic. Here, the residual frequency noise PSD is mostly limited by the Rayleigh scattering-limited relative intensity noise (RIN) of the fiber link at $2f_{AO}$, where the conversion of the Rayleigh scattering RIN to the frequency noise PSD scales with the fiber delay (τ) [Cahill et al, Opt. Express 23, 6400 (2015)]. On the other hand, the impact of timing detection sensitivity scaling to the frequency noise PSD inversely scales with the square of the fiber length ($1/\tau^2$) because the timing detection sensitivity of the fiber link scales with the fiber delay. As a result, when the Rayleigh scattering RIN scaling and timing detection sensitivity scaling are combined, the resulting frequency noise PSD inversely scales with the delay ($S_\xi(f) \sim 1/\tau$) as shown in Figure S8.

Revisions made: As the reason for the limited noise performance, we added “the Rayleigh scattering-limited intensity noise to the frequency noise conversion⁵⁴” in Lines 286-287 (page 13) with new ref. 54 [Cahill et al, Opt. Express 23, 6400 (2015)]. More detailed explanation outlined above is added in Supplementary Information (in page 9): “The main reason for the residual white noise floor (in frequency noise PSD) is the Rayleigh scattering RIN. While the Rayleigh scattering PSD scales with the delay length¹⁰, the enhanced detection sensitivity (which scales to the delay length) improves the phase noise PSD by the square of the delay length. As a result, when the Rayleigh scattering RIN scaling and timing detection sensitivity scaling are combined, the resulting frequency noise PSD floor inversely scales with the fibre delay length.”

Reviewer 2

Thanks for the detailed response. The authors' main argument is that they achieve a higher locking bandwidth (100 kHz) by combining a voltage-controlled oscillator and an acousto-optic frequency shifter (AOFS), instead of the use of a PZT stretcher (10 kHz) in ref. 24.

*This improvement is a very straightforward idea and merely a technical improvement. In fact, similar laser frequency control by using voltage-controlled oscillators and a frequency shifter has been described in *Phys. Rev. Lett.* 121, 063902, 2018. In addition, there are many techniques to control laser frequency at microsecond timescales.*

*In conclusion, this doesn't convince me that the revised manuscript has enough novelty. I think readers of the *Nat. Commun.* would expect more scientific advance, not just an improvement of locking bandwidth. This work is an extension from their previous work that is cited as Ref. 24.*

We respectfully disagree with the reviewer's opinion on the novelty of our work. The major contribution of our study is to provide a compelling approach for stabilizing the timing of ultrahigh-Q micro-combs on a simple, compact and robust platform, allowing chip-scale micro-combs to be used to their full potential. To our knowledge, very few studies have sought for such a compact yet high-performance micro-comb stabilization method, and we presented a novel approach with one of the lowest microwave phase noise performances for micro-comb systems. We remain confident that our result represents a major breakthrough in micro-comb and microwave photonics fields.

In addition to the overall impact and novelty aspects outlined above, we'd like to address the reviewer's comments on the frequency control methods. While the method shown in *Phys. Rev. Lett.* 121, 063902 (2018) appears to use a similar frequency control approach, a closer look reveals that it is different from our approach and, as a result, could not demonstrate broadband stabilization. There are many techniques for controlling the laser frequency at a microsecond timescale, as the reviewer pointed out, but there has been no work that accomplished broadband frequency modulation and stabilization for ultrahigh-Q microresonators. It has been particularly challenging because such ultrahigh-Q microresonators require continuous soliton-mode stabilization against thermal destabilization. As a result, broadband locking of repetition-rate in ultrahigh-Q microresonators has not been straightforward, and our result in this study represents one of the major advances in ultrahigh-Q microcombs compared to our previous work (ref. 24) and other laser frequency control technique studies.

Reviewer 3

*The authors have correctly answered my questions and added the supplementary information required to better understand these points. As I said in my first review, it is a new and very interesting work that must be accepted for publication in *Nature Com.**

We thank the reviewer for recognizing the importance of our work and supporting the publication.